

# Satellite soil moisture data assimilation impacts on modeling weather variables and ozone in the southeastern US - Part 2: Sensitivity to dry deposition parameterizations

Min Huang[1,a], James H. Crawford[2], Gregory R. Carmichael[3], Kevin W. Bowman[4], Sujay V. Kumar[5], and Colm Sweeney[6]

[1]College of Science, George Mason University, Fairfax, VA, USA
[2]NASA Langley Research Center, Hampton, VA, USA
[3]College of Engineering, The University of Iowa, Iowa City, IA, USA
[4]Jet Propulsion Laboratory, California Institute of Technology, Pasadena, CA, USA
[5]NASA Goddard Space Flight Center, Greenbelt, MD, USA
[6]NOAA Earth System Research Laboratory Global Monitoring Division, Boulder, CO, USA
[a]Now also visiting: National Centers for Environmental Prediction, College Park, MD, USA

*Correspondence to*: Min Huang (mhuang10@gmu.edu)

**Abstract.** Ozone ($O_3$) dry deposition is a major $O_3$ sink. Realistically representing this process in models is important for accurately simulating $O_3$ concentrations and exceedances, as well as assessing the $O_3$ impacts on human and ecosystem health. As a follow-up study of Huang et al. (2021), soil moisture (SM) data from NASA's Soil Moisture Active Passive mission are assimilated into the Noah-Multiparameterization land surface model within the NASA Land Information System framework, semicoupled with Weather Research and Forecasting model with online Chemistry regional-scale simulations covering the southeastern US. Major changes in the used modeling system include enabling the dynamic vegetation option, adding the irrigation process, and updating the $C_H$ (i.e., surface exchange coefficient for heat) scheme. Two different dry deposition schemes are implemented, i.e., the Wesely scheme and a "dynamic" scheme, in the latter of which dry deposition parameterization is coupled with photosynthesis and vegetation dynamics. It is demonstrated that, when the "dynamic" scheme is applied, the modeled $O_3$ dry deposition velocities as well as the total, stomatal and cuticular $O_3$ fluxes are overall larger and 2–3 times more sensitive to the SM changes due to the data assimilation (DA). We also highlight that, the configuration of the SM factor controlling stomatal resistance (i.e., the β factor which presents dependencies on soil type and hydrological regime) can strongly affect the quantitative results. Referring to multiple observation and observation-derived evaluation datasets, which may be associated with variable extents of uncertainty, the model performance of vegetation, surface fluxes, weather, and surface $O_3$ concentrations, shows mixed responses to the DA, some of which display land cover dependency. Finally, using model-derived concentration- and flux-based policy relevant $O_3$ metrics as well as their matching exposure-response functions, the relative biomass/crop yield losses for several types of vegetation/crops are estimated to be within a wide range below 20%. Their sensitivities to the model's dry deposition scheme and the implementation of SM DA are discussed.



# 1 Introduction

Ground-level ozone ($O_3$) is a regulated secondary air pollutant harmful to human and ecosystem health (Fleming et al., 2018; Mills et al., 2018a,b). It is closely connected with $O_3$ at higher altitudes where $O_3$ plays a more important role in the Earth's climate system. To better protect human health and public welfare, in 2015, the US primary and secondary National Ambient Air Quality Standards were lowered from 75 ppbv to 70 ppbv, in the format of daily maximum 8-h average (MDA8). Several other $O_3$-exposure based metrics have also been applied or/and proposed to assess $O_3$ impacts on vegetation, such as the

accumulated $O_3$ exposure over given thresholds (e.g., SUM40, SUM60, and AOT40), the averaged $O_3$ exposure during daylight hours (e.g., M7 and M12), and the sigmoidal-weighted W126 cumulative exposure (e.g., Fredericksen et al., 1996; van Dingenen et al., 2009; Hemispheric Transport of Air Pollution, 2010, and references therein; Avnery et al., 2011; Hollaway et al., 2012; Huang et al., 2013; Lapina et al., 2014; Mills et al., 2007, 2018a,b). To help comply with the ever-tightening air quality standards, an improved understanding of the individual processes affecting (near-)surface $O_3$

concentrations and exceedances is demanded. Many $O_3$-related processes are highly sensitive to environmental and/or biophysical conditions (Huang et al., 2021, and references therein). These $O_3$-related processes include dry deposition of $O_3$ and its precursors, which is a major sink for near-surface $O_3$ and depends on dry deposition velocities ($V_d$) and the deposited chemicals' concentrations. As recognized in numerous studies, accurately estimating dry deposition fluxes is critical to understanding $O_3$ budgets and exceedances in the past, present, and future (e.g., Stevenson et al., 2006; Griffiths et al.,

2021); it could also contribute to a more reasonable assessment of the $O_3$ impacts on vegetation (e.g., Mills et al., 2011; Lombardozzi et al., 2015; Mills et al., 2018b; Ducker et al., 2018; Ronan et al., 2020), which is relevant to the budgets of other greenhouse gases as well.

Ozone uptake by plants is generally higher in warm/growing seasons and during the daytime when $O_3$ concentrations and $V_d$

values peak. As introduced in Huang et al. (2021) as well as references therein, over the land, surface resistance $R_c$, which is composed of stomatal–mesophyll ($R_s$–$R_m$), cuticular ($R_{lu}$), in-canopy, and ground resistance terms, often exerts the strongest effects on the magnitude and variability of $V_d$. $V_d$ also includes the aerodynamic resistance ($R_a$) and quasi-laminar sublayer resistance ($R_b$) terms.

Soil moisture (SM) and its variability impact $V_d$ in the following ways: 1) SM can play a key role in controlling the opening and closing of plants' stomata as well as the mesophyll functioning (Egea et al., 2011; Baillie and Fleming, 2019), and thus it can directly affect the $R_s$ and $R_m$ terms of $V_d$; 2) SM is closely linked with vegetation attributes, such as the growing-season aboveground biomass, which is often expressed as leaf area index (LAI) or vegetation optical depth (VOD) and controls the stomatal and cuticular uptake of $O_3$-related species; and 3) SM as well as vegetation conditions can affect multiple $V_d$ terms

through its interactions with other environmental conditions (e.g., temperatures, radiation, precipitation and humidity fields) that modulate these $V_d$ terms, and such effects are generally stronger over transitional climate zones located between dry and



wet climates. It is expected that the impacts of SM on $V_d$ and atmospheric states through the above-mentioned pathways will continue to increase as the occurrence and severity of droughts, some of which are characterized by surface and/or column-averaged SM deficits, are projected to increase over many US regions under warmer future environments (Intergovernmental

Panel on Climate Change, 2021). Better understanding the potentially enhanced SM dependency of dry deposition and weather conditions under the changing climate is important because $O_3$ stress, together with heat, water, as well as other stresses, can pose more complex threats to plant health than single stress alone (Out-Larbi et al., 2020).

Chemical transport models have long been used to estimate $V_d$ values and their responses to climate change. In the widely-

used, empirical Wesely scheme (Wesely, 1989), $V_d$ is sensitive to only a few meteorological variables, with SM and plants' physiological effects ignored. In previous studies, Wesely scheme based $V_d$ fluxes as well as their various terms from different global and regional chemistry modeling systems were intercompared or/and evaluated with $V_d$ and $R_s$ observations from sparsely-distributed sites (e.g., Val Martin et al., 2014; Hardacre et al., 2015; Silva and Heald, 2018; Wu et al., 2018; Lin et al., 2019; Clifton et al., 2020) in terms of their magnitude and variability. Even when similar (Wesely and Wesely-like)

$V_d$ schemes were applied, various models behaved differently in calculating $V_d$, reflecting the impacts of land use/land cover (LULC) and meteorological fields which depend on the individual models' configurations (e.g., scales, inputs). Large model-model and model-observation discrepancies (i.e., by a factor of 2 or more) have been found in places, suggesting the strong needs of diagnosing and addressing issues in the models' configurations and $V_d$ parameterizations.

Revised or alternative dry deposition schemes have been applied in an increasing number of global- and regional-scale modeling studies. In some of these works, $R_s$ is calculated based on multiplicative algorithms in which empirical maximum stomatal conductance is adjusted by multiple factors, including water availability and vegetation attributes (e.g., Anav et al., 2018; Falk and Søvde Haslerud, 2019). In others, $V_d$ calculations are coupled with photosynthesis and vegetation phenology (e.g., Val Martin et al., 2014; Lin et al., 2019; Clifton et al., 2020), which in this paper are frequently referred as "dynamic"

schemes. Such types of modifications have been the recommended directions for improving the estimates of $V_d$ as well as the $V_d$ and $O_3$ responses to climate change, in that they have been demonstrated to be capable of enhancing the dynamics of the modeled $V_d$ and reducing their systematic biases. However, results based on such updated $V_d$ schemes are still associated with variable extents of uncertainty due to limitations in model parameterizations (related to structures, empirical parameters and stress functions) or/and configurations. In some existing works that applied the "dynamic" schemes, such uncertainty

was quantified and addressed by simply scaling the fluxes resulting from the "dynamic" schemes towards flux measurements available at very limited locations during non-recent time periods (e.g., Val Martin et al., 2014). These types of modified $V_d$ schemes still require further investigations and optimizations, which can be approached by: 1) quantifying the sensitivities of process-based model variables to SM and other environmental or/and biophysical variables for various LULC and soil types; 2) improving model representations of processes central to SM states and land-atmosphere interactions, such as including

irrigation and other human activities, tuning surface exchange coefficient ($C_H$) scheme in land surface models (LSMs), and



using available observations to constrain (some of) the model land variables; and 3) including a wide range of observations and/or observation-derived carbon, water, and energy fluxes as well as vegetation states in model evaluation for broad geographical regions. Furthermore, it is important to explicitly connect the progress in dry deposition modeling with the impact assessments of $O_3$ and other air pollutants on ecosystem health, productivity, and diversity.


A regional-scale land modeling and SM data assimilation (DA) framework coupled with weather and atmospheric chemistry modeling by the Weather Research and Forecasting model with online Chemistry (WRF-Chem) is implemented in this work. Using this tool, we quantify and discuss the responses of $V_d$ and its key components as well as $O_3$ concentrations and plant uptake to SM changes due to the DA, for different soil texture, LULC and crop types. The central parts of this work rely on

the Noah-Multiparameterization (MP, Niu et al., 2011) LSM with dynamic vegetation that enables the implementation of a modified "dynamic" dry deposition scheme. With this modified scheme, both the indirect (i.e., via changing weather and vegetation fields) and direct effects of SM on dry deposition are considered in this modeling system. Results based on this modified and the WRF-Chem default Wesely schemes are compared and evaluated with independent datasets. As an extended work of Huang et al. (2021), here we continue to focus on the southeastern US during summer 2016 for which

period prior Noah- and Wesely-based model calculations were conducted and aircraft observations are available. This manuscript introduces the applied two dry deposition schemes in Section 2. It then presents results from this Noah-MP based modeling system, in comparison with those from Huang et al. (2021) (Sections 3.1–3.2). Discussions on $O_3$ concentrations and fluxes based on all related WRF-Chem simulations are also connected with the assessment of $O_3$ impacts on societies, ecosystem health, and crop yield (Section 3.3). Summary and suggestions on future directions are provided in Section 4.

**2 Methods**

**2.1 Modeling and DA experiments design**

The modeling tools and DA experiment design of this study were largely consistent with the Huang et al. (2021) study: we conducted model simulations over the southeastern US in a semi-coupled Land Information System (LIS)/WRF-Chem system without and with the assimilation of the enhanced SM retrievals from NASA's Soil Moisture Active Passive (SMAP)

mission. Two dry deposition schemes were applied in cases without and with the SM DA. The model domain, horizontal and vertical resolutions, atmospheric/land initialization and SM DA methods were adapted from our previous study based on the Noah LSM. Major model input datasets, physics and chemistry schemes were kept similar as before except a few aspects relevant to the upgrade of LSM from Noah to Noah-MP (version 3.6) and the implementation of an irrigation scheme to be introduced in Section 2.2.


Same as in Huang et al. (2021), the LULC and soil texture type inputs of our coupled modeling system were based on the International Geosphere-Biosphere Programme-modified Moderate Resolution Imaging Spectroradiometer and the State Soil





Geographic datasets, respectively. Crop type data from Monfreda et al. (2008) were used in the irrigation scheme and the assessment of the $O_3$ impacts on vegetation (Figure 1b), which are roughly consistent with the 2016 records from the US

Department of Agriculture National Agricultural Statistics Service for several major crops such as maize, soybean and wheat (https://nassgeodata.gmu.edu/CropScape, lass access: 8 November 2021). In Section 3 of this paper, model results are summarized and/or discussed by groups of grid-dominant LULC and soil type that are shown in Figure 1(a;d). The original 20 LULC types were grouped into urban and non-urban areas, and for vegetation-dominant areas, into forests, croplands, and shrub/grasslands, following the criteria introduced in Table S1. The grid-dominant LULC groups for vegetated regions used

in our analysis are vastly similar to independently-developed data products, e.g.: a dataset derived from the European Space Agency–Climate Change Initiative Land Cover project (https://gwis.jrc.ec.europa.eu/apps/country.profile/overview/USA, lass access: 8 November 2021), and the 2016 National Land Cover Database. Urban-dominant grid cells are well aligned with dense population areas (Figure 1c) based on the Gridded Population of the World version 4.11 (NASA Socioeconomic Data and Applications Center, 2018). Grid-scale discrepancies exist between the used LULC input and independent LULC

products, which, however, are not anticipated to considerably impact the results averaged by LULC groups. Three groups of soil are highlighted, namely sand/loamy sand, loam and clay. The original sand and loamy sand categories are combined because of their high sand fractions (http://www.soilinfo.psu.edu/index.cgi?soil_data&conus&data_cov&fract&methods, lass access: 10 December 2021).

## 2.2 Physics and configurations of the Noah-MP LSM

The Noah-MP LSM includes a number of improvements from Noah, and one of the enhanced features in Noah-MP is that it contains a separate canopy layer that explicitly computes photosynthetically active radiation, canopy temperature, and related energy, water, and carbon fluxes so that it facilies a dynamic vegetation model. A modified two-stream radiation transfer scheme was used to compute fractions of sunlit and shaded leaves and their absorbed solar radiation. The Ball-Berry type of $R_s$ scheme (e.g., Ball et al., 1987) was applied as required by the dynamic vegetation option. When this option is

used, green vegetation fraction (GVF) does not come from an input dataset as in Huang et al. (2021) but is related to LAI based on (1):

$$GVF = 1 - e^{-0.52LAI} \qquad (1)$$

Niyogi and Raman (1997) concluded that Ball-Berry along with two other physiological schemes, performed better on $R_s$ than the multiplicative Jarvis type which has been frequently used with the prescribed vegetation option. Specifically, it

helps better capture the variance in $R_s$ and is more responsive to environmental changes. As described in Appendix B of Niu et al. (2011), this scheme relates stomatal resistance $r_{s,i}$ of sunlit and shaded leaves $i$ to the photosynthesis rates ($A_i$) per unit LAI of sunlit and shaded leaves $i$ separately:

$$\frac{1}{r_{s,i}} = m \frac{A_i}{C_{air}} \frac{e_{air}}{e_{sat\,(TV)}} P_{air} + g_{min} \qquad (2)$$





where $C_{air}$ is $CO_2$ concentration at the leaf surface. For our study period this was set at 400 ppmv according to the median

value of Atmospheric Carbon and Transport (ACT)-America B-200 aircraft near-surface (i.e., >900 hPa) $CO_2$ observations,

which is close to the global monthly-mean $CO_2$ concentrations in August 2016

(https://gml.noaa.gov/webdata/ccgg/trends/co2/co2_mm_gl.txt, lass access: 8 November 2021); $TV$, $P_{air}$, $e_{air}$, and $e_{sat}(TV)$ are

canopy temperature, surface air pressure, vapor pressure at the leaf surface, and saturation vapor pressure inside leaf,

respectively; $g_{min}$ and $m$ are land cover dependent empirical parameters. $A_i$ is determined by equations (3)–(6):

$$A_i = I_{gs} \min (A_c, A_{L,i}, A_s) \qquad (3)$$

$$A_c = \frac{(c_i - c_{cp})V_{max}}{c_i + K_c(1 + \frac{o_i}{K_o})} \qquad (4)$$

$$A_{L,i} = \frac{(c_i - c_{cp})4.6\alpha PAR_i}{c_i + 2c_{cp}} \qquad (5)$$

$$A_s = 0.5 V_{max} \qquad (6)$$

where $I_{gs}$ is a TV-dependent growing season index, $A_C$, $A_{L,i}$, and $A_S$ are carboxylase-limited, light-limited, and export-limited

photosynthesis rates per unit LAI, respectively; $c_i$ and $o_i$ are $CO_2$ concentrations inside leaf cavity which is about 0.7 times of

the atmospheric $CO_2$ concentration and atmospheric $O_2$ concentration, respectively. $PAR$ represents the photosynthetically

active radiation per unit LAI. $c_{cp}$ is the $CO_2$ compensation point and it equals to $0.5\frac{K_c}{K_o}0.21o_i$, where $K_c$ and $K_o$ are the

Michaelis-Menton constants for $CO_2$ and $O_2$, respectively, varying with TV; $\alpha$ is the quantum efficiency.

$V_{max}$ represents the maximum rate of carboxylation, expressed as:

$$V_{max} = V_{max25}\alpha_{vmax}^{\frac{TV-25}{10}}f(N)f(TV)\beta \qquad (7)$$

where $V_{max}$ is maximum carboxylation rate at 25°C; $f(TV)$ is a function that mimics thermal breakdown of metabolic

processes; $f(N)$ is a foliage nitrogen factor; and β is the SM factor controlling $R_s$, which presents strong dependencies on soil

type and hydrological regime. In this study model results based on the Noah and the Community Land Model (CLM) types

of β schemes are compared (Table 1), the latter of which is known to often result in sharper and narrower ranges of variation

with SM than the former does.

Other Noah-MP configurations include: the three-layer snowpack physics and the CLASS snow surface albedo; the Jordan

scheme for partitioning precipitation into rainfall and snowfall; the Niu-Yang-2006 frozen soil permeability and supercooled

liquid water option; the Simple Groundwater Model runoff scheme; and the Monin-Obukhov $C_H$ scheme, which, unlike

Noah's default Chen97 scheme, accounts for the zero-displacement height. Being affected by stability correction and

additional effects of planetary boundary layer height on friction velocity, it is likely that the Monin-Obukhov scheme can

result in either weaker or greater $C_H$ (i.e., less or more efficient ventilation of the land surface) than the Chen97 scheme

during the daytime in summer (Niu et al., 2011; Yang et al., 2011).




Irrigation process was included in all Noah-MP based simulations in this study. The benefit of including irrigation relies on the choice and parameterization of the irrigation scheme, as well as the LSM model's inputs (Lawston et al., 2015). Sprinkler scheme was chosen as it was reported as the prevalent irrigation method in 2015 across the US and many of the states within our model domain (Dieter et al., 2018). Irrigation was triggered over irrigated land in growing season within local morning times (6–10 am) when rootzone SM drops below 50% of the soil field capacity. The irrigated land was determined by the

model's LULC input and irrigation intensity information in Salmon et al. (2015), and the rootzone area was derived from the maximum root depth, which varies by crop type and GVF.

**2.3 Wesely and dynamic O₃ dry deposition schemes**

Dry deposition velocity $V_d$ is estimated based on the resistance analogy approach:

$$V_d = \frac{1}{[R_a+R_b+R_c]} \tag{8}$$

$R_a$ and $R_b$ are aerodynamic resistance and quasi-laminar sublayer resistance, respectively, sensitive to surface properties such as surface roughness. Over the land, surface resistance $R_c$, the major component of $V_d$, is classified into stomatal–mesophyll resistance ($R_s$–$R_m$), cuticular resistance ($R_{lu}$), in-canopy resistance ($R_{dc}$ and $R_{cl}$), and ground resistance ($R_{ac}$ and $R_{gs}$):

$$R_c = \frac{1}{\left[\frac{1}{R_s+R_m}+\frac{1}{R_{lu}}+\frac{1}{R_{dc}+R_{cl}}+\frac{1}{R_{ac}+R_{gs}}\right]} \tag{9}$$

where $R_{dc}$ is resistance for gas-phase transfer affected by buoyant convection in the canopy when sunlight heats the (near-)surface, $R_{cl}$ is resistance for leaves, twigs, bark, and others in the lower canopy, $R_{ac}$ is resistance for transfer that depends mostly on canopy structure, and $R_{gs}$ is resistance for soil, leaf litter, snow, and others at the ground surface.

Two deposition schemes, namely the Wesely and a dynamic scheme, were applied in this study, in which $R_s$ and $R_{lu}$ are

treated differently. In the Wesely scheme, $R_s$ and $R_{lu}$ are calculated based on (10) and (11):

$$R_s = \begin{cases} r_s\left\{1+\left[\frac{200}{G+0.1}\right]^2\right\}\left\{\frac{400}{T_s(40-T_s)}\right\}\frac{D_{H_2O}}{D_x}, & 0\,°C \le T_s \le 40\,°C \\ \sim9999, & >40\,°C \text{ or } T_s < 0\,°C \end{cases} \tag{10}$$

$$R_{lu} = \frac{r_{lu}}{10^{-5}H+f_0} + 1000e^{-T_s-4}, \text{ for dry surfaces according to humidity and precipitation fields} \tag{11}$$

Where $r_s$ and $r_{lu}$ are LULC- and season-dependent constants subject to uncertainty; $G$ and $T_s$ are radiation and surface temperature, respectively, whose definitions are different than those of $PAR$ and $TV$ in equations (2)–(7); $D_{H_2O}$ and $D_x$ are

molecular diffusivities for water vapor and trace gas $x$ (e.g., O₃), respectively; $H$, which is sensitive to surface temperature, represents the Henry's law constant for the focused trace gas; and $f_0$ is a reactivity factor for oxidation. The Wesely-scheme related results new from this study and Huang et al. (2021) are compared (Table 1).



As expressed in equations (12), in the dynamic scheme, $R_s$ used in dry deposition modeling was taken from $R_s$ calculated
from Noah-MP's dynamic vegetation model, and thus considers the physiological process of leaf responses to photosynthesis
rate, humidity and $CO_2$ concentrations. The direct effects of SM, as reflected in the β formula, as well as other environmental
variables, are included in this method, and this work quantifies the impact of the β factor configurations in Noah-MP (Table
1) on the dynamic-scheme-related results.

$$R_s = \left( \frac{r_{s,sunlit}L_{sunlit} + r_{s,shaded}L_{shaded}}{LAI} \right) \frac{D_{H_2O}}{D_x} \tag{12}$$

where $r_{s,sunlit}$ and $r_{s,shaded}$ are computed based on equations (2)–(7), $L_{sunlit}$ and $L_{shaded}$ are proportional to the sunlit and
shaded fractions of canopy, respectively, calculated based on the modified two-stream radiation transfer scheme.

In the dynamic scheme, $R_{lu}$ for dry surfaces is modified from the Wesely formula by considering its LAI dependency:

$$R_{lu} = \frac{r_{lu}}{LAI \times (10^{-5}H + f_0)} + 1000e^{-T_s - 4} \tag{13}$$

In both the Wesely and the dynamic schemes, $R_{dc}$ is sensitive to surface radiation, and $R_m$ is expressed as:

$$R_m = \frac{1}{\frac{H}{3000} + 100f_0} \tag{14}$$

Similar to the $R_{lu}$ calculations in equations (11) and (13), to approximate an effect that coldness sometimes reduces the
uptake, $1000e^{-T_s - 4}$ is added to LULC- and season-dependent constants to derive $R_{gs}$ and $R_{cl}$. It is worth mentioning that
the direct effects of water stress on mesophyll resistance have been recognized (e.g., Egea et al., 2011). Yet, in neither
scheme we applied, such effects have been incorporated into the $R_m$ formula as a part of the $V_d$ calculation.

## 2.4 Model evaluation, analysis, and O₃ impact assessments

For the cases listed in Table 1, we quantify the impacts of SM DA on the modeled SM, vegetation dynamics, surface fluxes
(i.e., gross primary productivity, GPP, which is integrated by LAI from $A$ in equations (2)–(3), energy fluxes and their
partitioning in the format of evaporative fraction, dry deposition flux and individual V$_d$ terms for O₃ particularly $R_s$ and $R_{lu}$
related), meteorological and surface O₃ fields during the 16–28 August 2016 period. The SM DA impacts on most of these
model fields are expressed as daily or/and daytime (~13:00–24:00 UTC) averaged absolute or relative changes referring to
the results from the no-DA cases. For O₃ dry deposition fluxes, we also conducted linear regression analyses to determine the
relationships between the relative flux changes versus the relative changes in column-averaged initial SM due to the DA.
Results of O₃ dry deposition fluxes and the regression analyses (i.e., slopes, correlation coefficient $r$ values, and $p$ values) are
summarized by grouped LULC types defined in Figure 1a.


A variety of data products were utilized in this study to assess the model performance in no-DA and DA cases. Many of
these evaluation datasets have been applied and introduced in detail in Huang et al. (2021), which are: 1) National Centers
for Environmental Prediction Global Surface Observational Weather Data as well as weather data collected onboard the
NASA B-200 aircraft during the ACT-America campaign; 2) hourly O₃ measurements at the US Environmental Protection



Agency Clean Air Status and Trends Network (CASTNET) and Air Quality System (AQS) sites; and 3) the 0.5°×0.5°, daily FLUXCOM latent and sensible heat fluxes. New evaluation datasets used in this work include: 1) the 9 km VOD retrievals from the enhanced SMAP product, which indicates the attenuation of microwave signals by vegetation, proportional to above-ground canopy biomass, and was used together with a 10-day average Copernicus Global Land Service GVF product to derive GVF for the focused 13-day period; 2) the daily GPP estimates from the 9 km SMAP level 4 carbon (L4C) product

version 6, developed based on the SMAP L4 surface (0–5 cm) and rootzone (0–100 cm) SM together with satellite LULC and vegetation datasets, which was supplemented by two independent GPP proxies (Whelan et al., 2020) of satellite-derived solar-induced chlorophyll fluorescence (SIF) data (Yu et al., 2019) and the Portable Flask Package (Sweeney et al., 2015) carbonyl sulfide (OCS) measurements collected onboard the B-200 and C-130 aircraft during the ACT-America campaign, and other airborne trace gas (e.g., benzene) measurements during this campaign were analyzed together with the OCS data to

help distinguish the influences of combustion sources from plant $CO_2$ uptake on the observed OCS distributions; and 3) $V_d$ data from selected CASTNET sites, estimated using a multilayer model (MLM, not supported by CASTNET as of 2017) version 3.0 which has known limitations and biases against eddy covariance flux measurements as well as $V_d$ estimated using other methods (e.g., Finkelstein et al., 2000; Saylor et al., 2014; Wu et al., 2018). The known limitations of MLM and how they may affect our model comparisons with the CASTNET $V_d$ data are discussed. Our $O_3$ dry deposition results are

also compared with eddy covariance measurements reported in independent works for similar climate or/and LULC types during other time periods.

This study also evaluates how the SM DA affected the assessments of surface $O_3$ impacts on human and ecosystem health. Specifically: 1) MDA8 $O_3$ fields over unban and nonurban regions were investigated linked to their respective population

ranges; and 2) LULC-specific $O_3$ stomatal uptake $F_s$ in the format of Phytotoxic Ozone Dose above the critical level of $y$ nmol m$^{-2}$ s$^{-1}$ (POD$_y$), as well as crop-specific AOT40, were evaluated based on equations (15) and (16).

$$\text{POD}_y \text{ (mmol m}^{-2}) = \sum\left[(F_s - y) \times \frac{3600}{10^6}\right], \text{ for hourly daytime stomatal uptake } F_s > y \text{ nmol m}^{-2} \text{ s}^{-1} \qquad (15)$$

$$\text{AOT40 (ppmh)} = \sum[(C - 0.04)] \text{ for hourly daytime } O_3 \text{ concentration } C > 0.04 \text{ ppmv} \qquad (16)$$

The calculated POD$_y$ and AOT40 were used to estimate the Relative Biomass Loss (RBL) or Relative Yield Loss (RYL) for

several types of vegetation or crops based on dose-response functions reported in literature (Table 2, Convention on Long-Range Transboundary Air Pollution, CLRTAP, 2017; Mills et al., 2007, 2018a). Our 13-day WRF-Chem model results were linearly-extrapolated to ~three months to derive the POD$_y$ and AOT40 fields. We focus on qualitatively interpreting the results and discussing their implications. The outcome from this analysis is also compared with the findings from several independent $O_3$ impact assessment studies for different time periods.




## 3 Results and discussions

### 3.1 Modeled SM and vegetation fields

Figure 2 compares the horizontal and vertical gradients of the modeled initial SM conditions from the Noah_D and CLM_D cases defined in Table 1. At the surface layer (0–10 cm belowground), both cases produced SM horizontal gradients that
resemble the Noah-based results presented in Huang et al. (2021). They are moderately correlated with the column-averaged SM fields ($r$=0.875 and 0.871, respectively), and the mean differences in column-averaged and surface SM from the Noah_D and CLM_D cases are 0.003 and -0.006 $m^3$ $m^{-3}$, respectively. Kumar et al. (2009) have found that, compared to other LSMs such as the Catchment (based on which the SMAP L4 datasets are produced), the 4-soil-layer Noah and 10-soil-layer CLM LSMs display successively weaker surface-subsurface coupling strengths, and the weakest coupling strength of CLM was
primarily attributed to its significantly larger number of soil layers. The slightly weaker surface-subsurface correlations in the CLM_D case than in the Noah_D from this work, both are based on a 4-soil-layer Noah-MP modeling system, indicate the minor role of the LSM physics, in particular the β factor scheme, in controling the vertical coupling strength of SM conditions.

The modeled SM fields from the Noah_D and CLM_D differ on grid scale, particularly in the subsurface zones (Figure 2a-b). For example, in sand-dominant regions that were experiencing drought conditions during this period (e.g., Florida and the Texas-Oklahoma border regions, where simulated SM is mostly under 0.2 $m^3$ $m^{-3}$), column-averaged SM values from the CLM_D case are notably smaller than those from the Noah_D case. These results contrast with those reported by Niu et al. (2011), in which cases Noah-MP with the CLM-type β factor consumed less soil water, resulting in smaller SM variability
than did the Noah-type β factor during drought periods. In their cases focusing on loam and clay soil that have higher wilting points when the CLM-type β factor scheme was applied, plant transpiration ceased to save soil water under drought conditions. Our results can be explained by the steeper CLM-type β–SM curve than the Noah-type β–SM curve for low SM, sand-dominant areas, as illustrated in Figure 3a of Niu et al. (2011). For such conditions, Noah-MP with the CLM-type β factor produces stronger evapotranspiration (ET) and consumes more soil water, resulting in drier soil. For wet regions
where SM values exceed 0.4 $m^3$ $m^{-3}$, such as Louisiana and Arkansas, the CLM- and Noah-type β values are close to 1.0 and insensitive to soil type and SM variations; therefore, SM and ET produced from the Noah_D and CLM_D cases do not diverge. These findings corroborate the conclusions by Yang et al. (2011) that the degree of the β impacts on the SM–ET relationship should depend on the soil type and hydrological regime, and they are important for understanding the vegetation and surface flux results to be presented in the later parts of this paper.

Referring to the SMAP SM data, in general, surface SM produced by the no-DA modeling systems show wet biases in non-forested regions and dry biases over the forests for the study period. These SMAP–model discrepancies were successfully reduced by the DA for all vegetated LULC groups (Figure S1), leading to overall slightly drier soil in DA-enabled





simulations. For both the Noah_D and CLM_D cases, the DA adjusted the modeled SM fields across the entire soil columns,

demonstrating that observational information at the surface was propagated into deep soil layers. The SM responses to the

DA as a function of soil layer from the Noah_D and CLM_D cases are roughly similar but different at small spatial scales,

which reflect the controls of the β factor scheme on the surface-subsurface coupling strengths of the used modeling/DA

system. With the SMAP DA enabled, the $r$ values between column-averaged and surface SM from the Noah_D and CLM_D

cases increased to 0.902 and 0.897, respectively.


The satellite-derived GVF fields (methods introduced in Figure S2 caption) transition from low-to-moderate (<0.6) to high

(>0.8) values from the western (mostly shrub/grasslands) to the central and eastern parts (forests/croplands dominant) of the

study region, and such spatial gradients are highly correlated with the SMAP VOD retrievals (Figure 3a;d). The Noah_D and

CLM_D cases both moderately well reproduced these spatial patterns. Major differences between these cases are found in

dry sandy regions, where, as discussed in previous paragraphs, more soil water was consumed for ET and plant growth in the

CLM_D case and therefore higher GVF values are given. The DA adjusted the modeled GVF and SM fields toward similar

directions, with the relative changes in GVF overall smaller. While the SM changes in the Noah_D and CLM_D cases are of

close magnitudes, GVF responded more strongly in the CLM_D case except for sandy regions. Referring to the satellite-

derived GVF fields which are also subject to large uncertainty, the modeled vegetation fields are more effectively improved

by the DA over sparsely vegetated regions such as the South-Central Plains. The DA also remarkably reduced the model–

satellite mismatches over some of the dense vegetation regions such as the southwestern Ohio. The likely degraded model

performance over certain dense vegetation areas can be partially explained by weaknesses related to SM-vegetation growth

feedbacks in the dynamic vegetation model parameterizations which need to be identified and addressed in future work. It is

also suggested that jointly assimilating satellite SM and vegetation phenology products such as the VOD retrievals needs to

be experimented which may maximize the positive DA impacts on multiple land variables and their atmospheric feedbacks.

### 3.2 Modeled fluxes and weather conditions

### 3.2.1 Carbon/energy fluxes and weather conditions

Figure 4 compares the spatial distributions of the period-mean WRF-Chem carbon and energy fluxes with SMAP L4C and

FLUXCOM products which contain observation information, and Table 3 summarizes WRF-Chem and observation-derived

flux results by three LULC groups. The observation-derived products indicate the highest GPP and evaporative fraction (EF

= daily latent heat/(daily latent heat + daily sensible heat)) over croplands. Without the DA, the Noah-MP related cases

outperformed the Noah related P1_W case on simulating EF, especially over shrub/grassland and cropland regions. This

indicates that, from Noah to Noah-MP, the multiple updates in LSM physics related to $R_s$, irrigation and $C_H$, are beneficial.

Larger GPP and EF values are found in CLM_D than in Noah_D, most of which match better with the SMAP L4C and

FLUXCOM data. The DA led to increased EF over shrub/grasslands in all model cases as well as over croplands in the



Noah_D case, bringing the model results closer to the FLUXCOM data. The EF values were unfavorably reduced by the DA in the CLM_D and P1_W cases over croplands and in all model cases over forests, reflecting the challenges of satellite SM DA over regions with dense vegetation and/or affected by human activities, which have also been reported in previous studies. For the Noah_D and CLM_D cases, this may also be due to the possibly degraded vegetation performance discussed

in Section 3.1. The modeled GPP in the CLM_D cases were lowered by the DA overall, which helped reduce the model–SMAP L4C discrepancies over forests and croplands. In the Noah_D case, GPP was improved by the DA over forests and (slightly) over shrub/grasslands. Based on the evaluation statistics, for this case, the CLM type of β factor scheme is shown slightly superior to the Noah type. Note that the quality of the SMAP L4C and FLUXCOM products may also be strongly LULC dependent, e.g., it has been known that the uncertainty of SMAP L4C data is generally larger for highly productive

plant functional types (Kimball et al., 2020). Such evaluation, therefore, has demonstrated the critical role of LULC type in understanding the model performance of carbon and energy fluxes and its responses to satellite SM DA.

Additional datasets were also utilized to help understand terrestrial carbon uptake, including satellite SIF and ACT-America aircraft OCS as well as its vertical gradients (Figure S3). Consistent with the SMAP L4C and WRF-Chem based results, the

largest SIF values are shown over croplands, especially maize and soybean fields in Illinois and Indiana, 2–3 times as high as those over shrub/grasslands in the South-Central Plains. Free-tropospheric (>2 km, above ground level, a.g.l.) OCS mixing ratios are mostly higher than those near the surface (≤2 km, a.g.l.), except for locations that may be strongly influenced by oceanic and anthropogenic combustion sources according to independent studies (Lennartz et al., 2017; Zumkehr et al., 2018) and other chemical tracers (e.g., benzene) measured onboard. The maximum OCS mixing ratios are higher than 550

pptv, and the OCS drawdowns (i.e., free-tropospheric minus near-surface concentrations) far exceed 60 pptv around the Lower Mississippi cropland regions and the Texas-Oklahoma border where soil was wet and likely an OCS source (Bunk et al., 2017). These values are much larger than those observed in summer 2004 over the eastern US (Campbell et al., 2008), indicating possible higher OCS emissions and stronger terrestrial carbon uptake in summer 2016 than in summer 2004.

In general, the modeled EF fields as well as their directions of changes due to the DA resemble those of latent heat flux and relative humidity (RH), which are opposite to those of sensible heat and surface temperatures (Figures 5 and S4). The model overall well reproduced the observed spatiotemporal variability of 2 m air temperature (T2) and RH, as well as FLUXCOM latent and sensible heat fluxes. The diagnostic 2 m weather fields and their responses to the DA strongly correlate with the model results at its surface level. The Noah-MP related cases reacted more strongly to the DA than the Noah-related cases,

with the responses in the CLM_D case larger than in the Noah_D case except for dry, sandy regions, which can be attributed to combined effects of the used $C_H$ and stomatal resistance schemes. It is important to note that diagnostic temperature and humidity variables are represented differently in Noah and Noah-MP and thus are not directly comparable. Specifically, in Noah, T2 is an explicit function of skin temperature, air density, specific heat of dry air at constant pressure, and 2 m surface exchange coefficient for heat, and 2 m specific humidity is a function of surface specific humidity, upward moisture flux at





the surface, air density and 2 m surface exchange coefficient for moisture; whereas in Noah-MP, they are expressed as functions of temperatures and water vapor for vegetated land and bare soil being weighed by their respective fractions. We therefore focus on quantitatively evaluating and intercomparing prognostic model weather variables (i.e., the model-level air temperature and humidity) against ACT-America aircraft observations (Figure 6). For air temperature, at all altitudes and near the surface (i.e., ≥800 hPa), the CLM_D case responded most strongly to the DA, and the DA-enabled CLM_D case

outperformed the Noah_D and P1_W cases. This performance is qualitatively consistent with the model's sensible heat performance referring to the FLUXCOM data. As for humidity, despite the most significant DA improvements in CLM_D, the Noah-MP related cases did not perform as well as the Noah related cases, which is also found in the model's latent heat performance in comparison with the FLUXCOM data. However, note that the model's humidity performance is more strongly related to that of $R_s$ and $V_d$ in the Noah-MP based cases via the direct impacts of humidity on $R_s$ calculations

(equation 2). The solar radiation fields from all model cases, which play vital roles in controlling the land-atmosphere exchanges of water and trace gases, do not differ dramatically and their responses to the DA are negligible (e.g., Figure 5g-l). This indicates that the DA impacts on the modeled surface fluxes resulted primarily from the changes in the modeled SM, humidity, surface/canopy temperatures, as well as vegetation fields.

### 3.2.2 Ozone dry deposition velocities and fluxes

Figure 7 presents the period-mean, daily-averaged $V_d$ and dry deposition flux $F_t$ (i.e., $V_d$ multiplied by concentration at the surface level, Wesely, 1989) for $O_3$ from all model cases, along with their responses to the SMAP DA. The daytime averages of these fields have similar spatial gradients but of larger magnitudes (not shown in figures). Table 3 summarizes for three LULC groups the daily- and daytime-averaged results. The modeled stomatal–mesophyll and cuticular conductances, as well as their diurnal variability are indicated in Figure 8. All model cases produced lower $V_d$ and $F_t$ values over shrub/grasslands

than over forests and croplands, qualitatively consistent with results from many existing model- and measurement-based studies. The results from Noah_W and P1_W, both of which are based on the same scheme (Wesely), are generally similar, with minor differences largely attributed to different surface temperature fields (Figures 5 and S4). The WRF-Chem modeled $V_d$ and $F_t$ fluxes were more strongly affected by the upgrade from the Wesely to the dynamic scheme: i.e., with the updated scheme, they show enhanced magnitudes, stronger spatial variability, as well as more intensive responses to the DA,

especially over forests and croplands. These results can be mainly explained by the fact that the stomatal–mesophyll and cuticular resistances in the dynamic scheme are sensitive to more environmental and biophysical variables, accounting for both the direct and indirect (i.e., via influencing the weather fields and plants' physiology) effects of SM on $V_d$. $V_d$ from the Noah_D and CLM_D cases, as well as its major term stomatal–mesophyll conductance, show strong correlations with the modeled GPP, latent heat, and EF fields which have been discussed in earlier sections. Comparing the cases that

implemented the CLM- and Noah-like β schemes, $O_3$-related fluxes resulting from the former configuration are of notably larger magnitude, spatial variability and absolute changes due to the DA. The SM impacts on the modeled $V_d$ and $F_t$ were further quantified using linear regression analyses between the relative changes in the modeled $O_3$ fluxes due to the DA



versus those in column-averaged SM initial conditions. All regression models yielded low $p$ values (i.e., $\ll 0.01$), suggesting good $\Delta V_d \sim \Delta SM$ and $\Delta F_t \sim \Delta SM$ relationships. The regression slopes are summarized in barplots (Figure 9) by three LULC

groups for all model cases in Table 1. For all LULC groups, the slopes based on the two cases that implemented the dynamic scheme are 2–3 times larger than those from the two cases using the Wesely scheme, and the slopes differ most strongly among cases over forests and croplands. The low $r$ values associated with several regression models (denoted by red "L"s in Figure 9) reflect the stronger nonlinear relationships between the changes in the studied $O_3$ fluxes and SM. These results emphasize the importance of better understanding and realistically representing in models the SM control on plants' stomatal

behaviors which regulate the land-atmosphere exchanges of water, energy, and trace gases. The earlier evaluation of the period-mean GPP and ET across the domain have demonstrated some advantages of using the CLM-like $\beta$ scheme, and that the DA more effectively improved the model performance in sparsely vegetated shrub/grassland regions. These conclusions are likely also applicable to the modeled $O_3$ dry deposition process, particularly its stomatal–mesophyll pathway.

In all no-DA and DA cases, the diurnal variability of $O_3$-related surface fluxes shows clear LULC dependency. Over the shrub/grassland and forests/croplands regions, the daytime averaged $V_d$ values are 24–31% and 35–50% higher than the 24 h mean, respectively, while the daytime averaged $F_t$ results are 40–50% and 42–63% higher than the 24 h mean, respectively (Table 3). Such $V_d$ diurnal cycles are a result of the strongest diurnal variability in stomatal–mesophyll conductance (i.e., its daytime mean values are ~twice as high as the 24 h mean for all LULC types) being balanced out by weak diurnal variability

associated with other $V_d$ terms. As the most diurnally variable $V_d$ component, stomatal–mesophyll conductance, on average, contribute less substantially to $V_d$ for shrub/grassland areas (24 h/daytime: up to ~30%/40%) than for forests/croplands (24 h/daytime: up to ~50%/66%), which helps explain the weaker diurnal variability in the modeled $V_d$ over shrub/grasslands. The stronger diurnal cycles in $F_t$ than in $V_d$ reflect the impacts of higher daytime $O_3$ surface concentrations used in the $F_t$ calculations. The DA did not dominantly intensify or dampen the diurnal cycles of these fluxes for any given grouped LULC

type. Whether the DA improved the estimated diurnal cycles of fluxes for various LULC types remains to be evaluated, which can benefit from independent observation-constrained flux products of broad spatial coverage and subdaily variability.

A detailed analysis was then conducted at two forest CASTNET sites with different soil types and hydrological regimes. The modeled $V_d$ and $F_t$ from various cases are compared with the operational MLM-based calculations produced at a Florida site

SUM156 and a Virginia site PED108 (Figure 10a-d; Table 5). The dominant soil types at these sites are sand and loam, and the column-averaged SM values from various model cases are approximately 0.15 and 0.20 $m^3$ $m^{-3}$, respectively. All datasets show that $V_d$ and $F_t$ sharply increase soon after sunrise, reaching their daily maxima in late morning or early afternoon. The slight declines in fluxes around midday based on some simulations can result from the water and heat stresses which cause stomata closures. The water stress starts to get relieved since the mid-afternoon at the SUM156 site under the influences of

convective precipitation whereas persists throughout the afternoon at the PED108 site (Figure 10e-f), which helps shape the slightly different afternoon flux dynamics at these two locations. Without the DA, at both sites, the highest daytime fluxes



were produced from the CLM_D case, followed by the Noah_D and Noah_W cases, which are 2–3 times as high as the MLM-estimated. The fluxes from all WRF-Chem cases during the nighttime are close, up to >80% lower than their daytime maxima, but still dramatically higher than the MLM-based results which are nearly zero. Our nighttime $V_d$ results are close

to flux observations at European forest sites during both dry and wet periods in the past decades (Lin et al., 2020). Wu et al. (2018) compared $V_d$ observations with model calculations based on the operational MLM, Wesely, and a dynamic scheme Noah-GEM, at a Canadian mixed forest site dominated by sand-like soil. Their diverse model results are qualitatively consistent with our findings at these two CASTNET sites. The remarkably lower $V_d$ values from the operational MLM calculations are attributed to its simplified approaches of calculating $R_a$ and $R_b$ using wind speed and direction, the empirical

approach of calculating $R_s$, and the lack of continuous, accurate model input data. Within the respective ranges of the modeled SM at these two sites, β factors based on the CLM-type scheme are both larger than those based on the Noah-type β scheme (referring to Niu et al., Figure 3), which helps explain the higher model fluxes from the CLM_D case than the Noah_D case without the DA. At the SUM156 site, despite the strongest SM decrease (~0.04 $m^3$ $m^{-3}$) by the DA in case CLM_D, the modeled fluxes responded least significantly to the DA, in part due to the flattened CLM-type SM–β curves in

contrast to the linear Noah-type SM–β function for sand within the 0.12–0.16 $m^3$ $m^{-3}$ SM range. At the PED108 site, the modeled SM values from all model cases were lowered by the DA by ~0.02 $m^3$ $m^{-3}$. The stronger flux reactions to the DA from the CLM_D case than those from the Noah_D case can be partially explained by the steep CLM-type SM–β curve versus the linear Noah-type SM–β relationship for loam within the 0.18–0.22 $m^3$ $m^{-3}$ SM range. Case studies at these two sites with the same type of LULC emphasize the importance of soil type and hydrological regimes for understanding SM

controls on dry deposition, which was often omitted or underdiscussed in previous dry deposition studies.

### 3.3 Policy-relevant $O_3$ metrics and implications for $O_3$ impact assessments

#### 3.3.1 MDA8 and implications for $O_3$ health impacts

Figure 11 illustrates the impacts of the choice of dry deposition scheme and SM DA on WRF-Chem modeled surface MDA8 $O_3$. During the study period, several warmer- and drier-than-normal Atlantic states experienced high MDA8 at times (i.e.,

≥60 ppbv, which can negatively affect lung function, and at ≥70 ppbv, cause respiratory symptoms and other adverse effects, Fleming et al., 2018, and references therein). Numerous populated urban centers reside in these areas. The levels of MDA8 are shown to be much lower (i.e., <40 ppbv) over the southern part of the domain, including several major urban/suburban regions such as the Texas Triangle, which was frequently influenced by passing cold fronts and tropical systems from the Gulf of Mexico.


All model cases reproduced these MDA8 spatial patterns moderately well. Referring to observations at AQS and CASTNET sites, their domain wide mean RMSEs all fall within 6–8.5 ppbv (Figure 11m). We first intercompare the MDA8 levels from all no-DA cases. Positive and negative differences between the results from Noah_W and P1_W, both of which implemented



the Wesely scheme, are almost equally distributed across the domain, with the MDA8 from the former case associated with
negligibly lower RMSEs (i.e., <0.02 ppbv on average) referring to AQS and CASTNET observations (Figure 11l-m). The
differences between these two cases are largely due to the impact of the chosen LSM on the model's meteorological fields,
particularly temperatures, which affected the simulations of various $O_3$-related processes including dry deposition. As Figure
11(j;k;m) show, replacing Wesely with the dynamic dry deposition scheme considerably lowered the calculated MDA8
levels in majority of the model grids, as well as their associated RMSEs (i.e., by >0.5 ppbv on average) relative to surface
observations. These reductions in MDA8 are of comparable magnitudes with those due to updating anthropogenic emissions
from the National Emission Inventory 2014 to 2016beta (Huang et al., 2021). Comparing the implementations of CLM- and
Noah-types of β schemes, the former led to stronger reductions in the modeled MDA8 fields and their associated uncertainty.
These results reflect the impacts of the faster $O_3$ removal via dry deposition in the dynamic scheme related cases, as well as
the different model meteorology. Our findings are qualitatively consistent with the conclusions from several global-scale
modeling experiments that compared the Wesely and dynamic schemes (e.g., Val Martin et al., 2014; Lin et al., 2019).

In all model cases, the DA reduced surface and subsurface SM in many of the grids, leading to enhanced MDA8 (Figure 11f-
i). The responses of the period-mean MDA8 to the DA from the Noah_W and P1_W cases are mostly within ±4 ppbv. When
the dynamic dry deposition scheme was applied, the modeled MDA8 responded several times more strongly to the DA (i.e.,
by up to 6 ppbv and 8 ppbv in the Noah_D and CLM_D cases, respectively), especially over nonurban regions where surface
MDA8 on average are several ppbv lower than in urban grids. In urban grids where population densities are ~25 times higher
than in nonurban grids (Figure 1c), the DA impacts on MDA8 reach 3–4 ppbv in places, under the controls of the local-to-
regional circulation patterns (Figure 12a;e). As the no-DA cases are positively biased against surface observations in many
places, corresponding to the DA-induced surface $O_3$ changes, the overall model performance of MDA8 was not improved, or
much degraded, by the DA. Over limited areas such as the South-Central Plains, the modeled MDA8 decreased due to the
DA by up to >2 ppbv, corresponding to improved performance. The no-DA and DA results based on different LSMs and dry
deposition schemes confirm that drier soil conditions exacerbate $O_3$ air pollution, which, together with heat stress, threatens
human health. Such $O_3$–SM relationships have also been demonstrated by Falk and Søvde Haslerud (2019) and Anav et al.
(2018) using other chemical transport models and multiplicative dry deposition schemes. Our Noah_W and P1_W related
results indicate the influences of SM on air quality via its feedbacks to weather; and results from the Noah_D and CLM_D
cases provide valuable information regarding both the indirect (i.e., via adjusting vegetation phenology and weather
conditions) and direct SM effects on $O_3$. The complex SM impacts on $O_3$ dry deposition as well as surface $O_3$ concentrations
based on the coupled photosynthesis-$R_s$ calculations rely heavily on the application of water stress function (β scheme), soil
properties and hydrological regime. The WRF-Chem results from this case indicate that, to more accurately simulate MDA8,
improving land DA must be combined with aggressive efforts to identify other sources of uncertainty in $O_3$ modeling (e.g.,
emissions, chemistry, and extra-regional pollution contributions) and reduce their negative impacts on model performance.



### 3.3.2 Implications for O₃ vegetation impact assessments using concentration- and flux-based metrics

Both O$_3$ flux- and concentration-based metrics have been applied to assess O$_3$ impacts on vegetation as well as the associated economic loss. Estimating the plants' stomatal O$_3$ uptake F$_s$ is the basis for constructing flux-based O$_3$ impact assessments.

Figure 13 illustrates the period-mean daytime F$_s$ fields based on all WRF-Chem no-DA cases as well as their responses to the SM DA. Box-and-whisker plots in Figure 12(b;f) summarize these results by three LULC groups. The averaged F$_s$ values for all three LULC groups exceed their respective critical levels (i.e., 1 nmol m$^{-2}$ s$^{-1}$ for forest and grasslands; and 3 nmol m$^{-2}$ s$^{-1}$ for crops). As a major contributor to O$_3$ dry deposition flux during the daytime, F$_s$ fields appear to be closely correlated in space and time with the surface humidity and flux fields (e.g., GPP, latent heat and EF, as well as V$_d$), which differ distinctly

from the surface O$_3$ concentration fields. For example, F$_s$ hotspots are shown over some low O$_3$ concentration areas including the humid, Lower Mississippi River regions, and the lowest F$_s$ values occur in certain high O$_3$ concentration regions strongly affected by urban pollution (e.g., Georgia) and pollution transport from upwind US states and/or the stratosphere (e.g., western Kansas and Oklahoma, as discussed in Huang et al., 2021). The changes in F$_s$ and surface O$_3$ concentrations due to the DA show opposite directions, i.e., drier soil enhances surface O$_3$ concentrations whereas slows

down the plants' stomatal O$_3$ uptake (Figures 11f-i and 13e-h). This comparison highlights how the choice of O$_3$ metrics can affect the assessment of O$_3$ vegetation impacts under the changing climate. As emphasized by Mills et al. (2018a) and Ronan et al. (2020), flux-based metrics have evident advantages over concentration-based metrics. To conduct reliable impact assessments using these flux-based metrics, accurate information on stomatal and non-stomatal fluxes as well as the various environmental and biophysical variables that they are sensitive to become increasingly important.


An assessment of O$_3$ vegetation impacts was conducted based on the results from various model cases and different metrics, namely POD$_y$ (where y is LULC-dependent critical level) and AOT40. For this demonstration, the 13-day model results were linearly extrapolated to ~three months. This also assumed similar DA adjustments to SM dynamics (driven by factors such as clouds/radiation, rainfall, and irrigation for cropland-dominant regions) at the time scale of ~three months. Based on the

known seasonal variability of surface O$_3$ and V$_d$ in the study region, the linearly scaled POD$_y$ and AOT40 values may have been overall underestimated. We therefore focus on discussing the results qualitatively and highlighting their implications for O$_3$ impact assessments using long-term records. Statistics of the derived POD$_y$ and AOT40 fields are summarized by O$_3$ sensitive LULC and crop types in Figure 12(c-d;g-h), and Figure 14 presents the estimated AOT40 fields as well as their responses to the SM DA for cropland-dominant grids. The highs and lows in AOT40-related results are found over maize

and wheat dominant fields, respectively. Among the three focused LULC types, the highest and lowest POD$_y$ values are estimated for forests and grasslands, respectively. Largely driven by daytime peak O$_3$ concentrations, the spatial variability and biases (referring to AQS and CASTNET observations) of the model-derived AOT40 fields, as well as their responses to the DA, match the MDA8-based (Figure 11). In contrast, the spatial variability of POD$_y$ and F$_s$ aligns well, so are their



responses to the DA. Both $POD_y$ and AOT40 reacted several times more intensively in the cases that implemented the
dynamic dry deposition scheme, especially the CLM_D case.

For selected LULC and crop types, the WRF-Chem derived $POD_y$ and AOT40 fields were used together with dose-response
functions in literature to evaluate the RBL/RYL due to $O_3$ exposure and uptake. As reported in Figure 12(c;g), with the SM
DA enabled, the mean RBLs based on Noah_D and CLM_D derived $POD_y$ are 0.06–0.10, 0.02–0.03, and 0.04–0.05 for
deciduous forest, grasslands and wheat, respectively, which are >33% lower than the Noah_W and P1_W based RBL
estimates. It is shown that, in response to the DA which lowered SM in many places, the Noah_W and P1_W based RBL
estimates did not drop as strongly as the Noah_D and CLM_D based, and even increased for grasslands and wheat. For
wheat, one of the most $O_3$-sensitive crops, the estimated RYL values based on $POD_y$ and AOT40 approaches differ by up to
a factor of 2–3, and the DA had contrasting effects on these estimates (Figure 12c-d;g-h). The $POD_y$- and AOT40-based
RYL values differ more significantly when the model-derived $POD_y$ and AOT40 fields came from the Noah_D and CLM_D
cases. Using the model-derived AOT40 and different AOT40 dose-response functions (Mills et al., 2007, 2018a, Table 2),
the estimated RYLs and their changes due to the DA are nonnegligible (Figure 12d;h). Our estimated RBL/RYL results for
various LULC and crop types mostly fall within the ranges reported in previous studies which applied model-derived $O_3$
metrics and dose-response functions (e.g., Avnery et al., 2011; Mills et al., 2007, 2018a). Our results emphasize that the
selected $O_3$ impact assessment metrics for various LULC/crop types and their matching dose-response functions, as well as
the model results used to derive the chosen $O_3$ metrics which are sensitive to dry deposition schemes and SM, all introduce
uncertainty to the estimated $O_3$ impacts on vegetation. The widely-used dose-response functions are considered appropriate
for studying North America and Europe, but they may not be applicable to other regions (Emberson et al., 2009). Therefore,
updating and developing dose-response relationships for a larger number of vegetation types in different regions of the world
are needed, which may require new experiments to be conducted. Yue and Unger (2014) and Lombardozzi et al. (2015) as
well as follow-on investigations parameterized the $O_3$ impacts on several types of vegetation using the relationships between
cumulative $O_3$ uptake and $O_3$ damage factors for photosynthesis and conductance from empirical and experimental studies.
Based on multi-decadal model simulations, they reported <20% changes of biomass, GPP, and energy fluxes due to $O_3$,
which are roughly consistent with our RBL/RYL results in Figure 12. Such approaches that dynamically assess the impacts
of $O_3$ along with other factors (e.g., non-$O_3$ pollutants and environmental stresses), as highlighted in Emberson et al. (2018),
will be considered in future work.

We note that, revising the dry deposition scheme and constraining the modeled SM fields with observations would not only
better be combined with adding $O_3$ injury to vegetation but also multistress impacts on biogenic emissions. Considering $O_3$
injury to vegetation would affect more evidently longer-term climate simulations via feedbacks to biomass, surface fluxes,
weather and weather-driven emissions. As for biogenic emissions, Figure S5 shows SM anomalies during the study period
determined by our Noah-MP modeling system as well as drought stress activity factor $\gamma_d$ estimated from $\beta$ of a multiyear,





independent CLM simulation by Jiang et al. (2018). Based on these, we estimate that, depending on soil type, hydrological regime, as well as β configurations, omitting the direct impacts of water stress on biogenic emissions, may have introduced
larger uncertainty to biogenic emission and $O_3$ modeling over several states experiencing drier-than-normal conditions, such as Tennessee, South Carolina, Alabama, and West Virginia. Quantitatively understanding the interplay between these processes and $O_3$ pollution levels is recommended for more accurate air quality modeling and $O_3$ impact assessments.

## 4 Summary and suggestions on future directions

This paper described a follow-up study of Huang et al. (2021). It presented how the choice of $O_3$ dry deposition scheme
affected our evaluation of SMAP SM DA impacts on coupled WRF-Chem modeling over the southeastern US in August 2016. In new Noah-MP LSM related simulations, two dry deposition schemes were implemented, namely the WRF-Chem default Wesely scheme and a dynamic scheme, in the latter of which the calculation of $V_d$ (particularly its stomatal and cuticular terms) were modified to be coupled with photosynthesis and vegetation phenology. We showed that dry deposition parameterizations significantly affected the modeled $O_3$ dry deposition process, as well as its response to the DA. Comparing
the no-DA cases, it was found that, when the dynamic scheme was applied, overall, the modeled $O_3$ dry deposition velocities and fluxes were larger and surface $O_3$ concentrations were lower. The modeled $O_3$ fluxes responded 2–3 times more strongly to the SM changes due to the DA, which can be mainly explained by the fact that both the direct and indirect (i.e., via influencing weather and vegetation fields) effects of SM on $O_3$ dry deposition modeling are considered in the dynamic scheme. Depending on soil type and hydrological regime, the selection of SM factor controlling $R_s$ (i.e., β factor, a key
variable representing the direct effects of SM on the modeled surface fluxes) scheme can strongly affect the quantitative results. The Wesely-scheme derived dry deposition results driven by meteorological fields from Noah-MP and Noah (from Huang et al., 2021) LSM based WRF-Chem simulations displayed much smaller differences than those due to updating the dry deposition parameterizations. While we note that accounting for physiological effects in dry deposition modeling can be beneficial, the Ball-Berry $R_s$ scheme applied in land surface and dry deposition modeling in this work needs to be compared
with other semi-empirical $R_s$ schemes, for a better understanding of their respective strengths and weaknesses. Alternative schemes include the Medlyn scheme which has been integrated into the CLM version 5. Model intercomparison efforts such as the ongoing AQMEII4 activity (Galmarini et al., 2021) can also help determine areas for improvement in commonly-used dry deposition modeling approaches for studying 2016 and other years, over North America and other regions of the world.

By analyzing the model responses to the SM DA from these various cases, we conclude that, in coupled modeling systems that consider the direct and indirect influences of SM on $O_3$ dry deposition, the accuracy of SM is particularly critical to dry deposition and $O_3$ modeling, as well as the scientific analyses and impact assessments based on model simulations. The usefulness of SM DA for improving the modeled state and flux variables was evaluated by multiple observation(-derived) data products. Referring to in situ measurements, key meteorological variables relevant to $V_d$ calculations such as surface





temperature and humidity are shown to be improved by the DA by up to ~9%. Referring to satellite(-derived) datasets which may be associated with high uncertainty, the model performance of vegetation phenology, GPP, as well as energy fluxes and their partitioning, showed mixed, LULC-dependent reactions to the DA. According to the evaluation statistics, for this case, the CLM type of $\beta$ factor scheme was slightly superior to the Noah type. The modeled carbon and energy fields as well as their DA-related changes, correlated strongly with the modeled $V_d$ fields, implying that the DA impacts on the accuracy of

$V_d$ were also possibly complicated which is difficult to verify due to the lack of high-accuracy, independent $V_d$ evaluation datasets. Observation(-derived) $V_d$ datasets covering diverse LULC types nested in broad geographical regions and through more recent periods are in strong need. In places, the likely ineffectiveness of SM DA on vegetation and surface fluxes can not only be attributed to the quality of satellite SM retrievals and the used DA approach as discussed in previous Noah LSM based DA experiments, but also shortcomings in the Noah-MP LSM and its dynamic vegetation scheme regarding its

surface-subsurface coupling and representation of SM-vegetation growth feedbacks. Continued efforts on advancing land measurement/retrieval skills, identifying and addressing deficits in LSMs as well as practicing multivariate land DA are recommended in future work.

This study also demonstrated that, model-driven assessments of $O_3$ impacts on human health and various types of vegetation

can be significantly affected by the applied $O_3$ dry deposition scheme, the implementation of land DA, the chosen $O_3$ metrics and their matching exposure-response functions. Various model cases showed that, the DA impacts on MDA8 were more evident in nonurban areas where the mean MDA8 was ~5 ppbv lower and the averaged population density is <1/25 of those in urban areas. Using concentration- and flux-based metrics AOT40 and $POD_y$, the mean RYLs of maize, soybean, and wheat fell within ranges of 0.03–0.07, 0.10–0.17, and 0.04–0.14, respectively. While the multiple no-DA and DA cases

helped us better understand the indirect or/and direct effects of SM on $O_3$ dry deposition process, which has important implications for $O_3$ impact assessments, it is recognized that, the DA often exacerbated the positive $O_3$ biases in free-running systems which has been a common issue shared by numerous regional and global models for this study region/season. It is necessary to combine land DA with efforts to identify and reduce other sources of uncertainty in $O_3$ modeling. These should include reasonably representing the impacts of $O_3$ along with other factors on vegetation as well as the direct impacts of

water stress on biogenic emissions of volatile organic compounds and nitrogen species.

**Code and data availability**

Dry deposition related updates to LIS/WRF-Chem since Huang et al. (2021) are undergoing reporting processes via NASA's New Technology Reporting System (https://invention.nasa.gov). Observations and observation-derived data sets used in this work but not in Huang et al. (2021) can be found at the following locations: https://doi.org/10.5067/L6C9EY1O8VIC

(Kimball et al., 2021), https://doi.org/10.7927/H49C6VHW (NASA Socioeconomic Data and Applications Center, 2018), https://www-air.larc.nasa.gov/cgi-bin/ArcView/actamerica.2016 (NASA, 2020, last access: 8 November 2021),





**Author contributions**

MH led the design and execution of the study as well as the paper writing, benefitting from discussions with JHC, GRC, KWB, and SVK, with the feedback from the Atmos. Chem. Phys. Editorial Board and reviewers for Huang et al. (2021) also accounted for. CS contributed to data collection during the ACT-America campaign. All authors helped finalize the paper.

**Competing interests**

The authors declare that they have no conflict of interest.

**Acknowledgements**

We acknowledge NASA SUSMAP sponsorship, as well as ACT-America Science Team and NASA's high-end computing systems and services at Ames and Goddard. We thank Jiang et al. (2018) for developing the $\gamma_d$ dataset shown in Figure S5. We also greatly appreciate active and relevant discussions with multiple colleagues from the Air Quality Model Evaluation International Initiative 4 and the Tropospheric Ozone Assessment Report II communities during and after recent conferences 660 and workshops, particularly: C. Hogrefe, J. Pleim, P. Makar, L. Emberson, B. Sinha, D. Lombardozzi, O. Clifton, L. Emmons, T. Emmerichs, and D. Taraborrelli.

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

**Tables**

**Table 1: Model cases and their configurations relevant to the discussions of this study.**

| Case name | Land surface model | Stomatal resistance scheme | SM factor controlling $R_s$ (β) | Surface exchange coefficient for heat ($C_H$) scheme | Irrigation scheme | Dry deposition scheme | Note |
|---|---|---|---|---|---|---|---|
| Noah_D | Noah-MP | Ball-Berry | Noah-type | Monin-Obukhov | Sprinkler | Dynamic | new in this study |
| CLM_D | Noah-MP | Ball-Berry | CLM-type | Monin-Obukhov | Sprinkler | Dynamic | |
| Noah_W | Noah-MP | Ball-Berry | Noah-type | Monin-Obukhov | Sprinkler | Wesely | |
| P1_W | Noah | Jarvis | Noah | Chen97 | not included | Wesely | from Part 1 |


**Table 2: Dose-response functions used to estimate the LULC- and crop-specific Relative Yield Losses (i.e., 1 - Relative Yield, RY) due to O₃ exposure and uptake, along with their references.**

| LULC type | Crop type | Dose-response function (references) | |
|---|---|---|---|
| | | Based on Phytotoxic Ozone Dose above the critical level y nmol m⁻² s⁻¹ (POD_y, in mmol m⁻²) | Based on AOT40 in ppmh |
| Deciduous Forest | / | RY= −0.0154 POD₁ + 1.012 (CLTRAP, 2017) | / |
| Grasslands | / | RY= −0.0074 POD₁ + 0.982 (CLTRAP, 2017) | / |
| Croplands | Maize | / | RY = −0.0036 AOT40 + 1.02 (Mills et al., 2007) |
| | Soybean | / | RY = −0.0116 AOT40 + 1.02 (Mills et al., 2007) |
| | Wheat | RY= −0.0064 POD₃ + 0.9756 (Mills et al., 2018a; CLTRAP, 2017) | RY = −0.0161 AOT40 + 0.99 (Mills et al., 2007) RY=−0.009 AOT40 + 0.969 (Mills et al., 2018a) |



**Table 3: Evaluation of daily-averaged WRF-Chem gross primary productivity and evaporative fraction.**

| Flux/weather variables (unit) | LULC type | Observation(-derived) | Model case | | | | | |
|---|---|---|---|---|---|---|---|---|
| | | | Noah_D | | CLM_D | | P1_W | |
| | | | No DA | DA | No DA | DA | No DA | DA |
| Gross primary productivity (g m$^{-2}$ d$^{-1}$) | forests | 7.39 | 7.88 | 7.08 | 9.06 | 6.94 | / | |
| | shrub/grass | 5.11 | 3.28 | 3.29 | 4.74 | 3.89 | | |
| | croplands | 8.94 | 7.64 | 7.40 | 9.77 | 8.13 | | |
| Evaporative fraction | forests | 0.75 | 0.65 | 0.60 | 0.67 | 0.60 | 0.66 | 0.63 |
| | shrub/grass | 0.67 | 0.53 | 0.58 | 0.57 | 0.61 | 0.48 | 0.48[a] |
| | croplands | 0.79 | 0.67 | 0.67[a] | 0.71 | 0.68 | 0.63 | 0.62 |

[a]The increases from no-DA cases, which led to improved model performance, are <0.005.

**Table 4: The 24 h and daytime mean O$_3$ deposition velocity (V$_d$) and flux (F$_t$) for three LULC groups.**

| LULC type/Model case | Noah_D | | CLM_D | | Noah_W | | P1_W | |
|---|---|---|---|---|---|---|---|---|
| | No DA | DA | No DA | DA | No DA | DA | No DA | DA |
| | 24 h mean V$_{d[ozone]}$ (cm s$^{-1}$) | | | | | | | |
| Forests | 0.64 | 0.56 | 0.68 | 0.51 | 0.54 | 0.53 | 0.49 | 0.48 |
| Shrub/Grass | 0.48 | 0.45 | 0.53 | 0.45 | 0.47 | 0.48 | 0.46 | 0.46 |
| Croplands | 0.62 | 0.54 | 0.67 | 0.54 | 0.58 | 0.58 | 0.56 | 0.56 |
| | 24 h mean F$_{t[ozone]}$ (nmol m$^{-2}$ s$^{-1}$) | | | | | | | |
| Forests | 7.11 | 6.38 | 7.47 | 6.35 | 6.31 | 6.24 | 5.75 | 5.68 |
| Shrub/Grass | 4.79 | 4.48 | 5.21 | 4.54 | 4.76 | 4.79 | 4.62 | 4.63 |
| Croplands | 6.90 | 6.11 | 7.39 | 6.06 | 6.69 | 6.64 | 6.44 | 6.42 |
| | Daytime mean V$_{d[ozone]}$ (cm s$^{-1}$) | | | | | | | |
| Forests | 0.94 | 0.80 | 1.02 | 0.71 | 0.79 | 0.77 | 0.70 | 0.69 |
| Shrub/Grass | 0.63 | 0.56 | 0.72 | 0.58 | 0.61 | 0.63 | 0.58 | 0.58 |
| Croplands | 0.88 | 0.74 | 0.99 | 0.73 | 0.83 | 0.83 | 0.80 | 0.79 |
| | Daytime mean F$_{t[ozone]}$ (nmol m$^{-2}$ s$^{-1}$) | | | | | | | |
| Forests | 11.51 | 10.04 | 12.25 | 8.99 | 10.05 | 9.93 | 9.04 | 8.88 |
| Shrub/Grass | 6.91 | 6.32 | 7.77 | 6.43 | 6.83 | 6.99 | 6.52 | 6.49 |
| Croplands | 10.99 | 9.42 | 12.04 | 9.31 | 10.61 | 10.57 | 10.17 | 10.07 |






**Table 5: Soil moisture and surface fluxes at two CASTNET sites shown in Figure 1d.**

| CASTNET sites (soil type; LULC type) | SUM156, Florida (sand; forest) | | PED108, Virginia (loam; forest) | |
|---|---|---|---|---|
| Modeled soil moisture initial condition, column-averaged ($m^3\ m^{-3}$) | No DA | DA | No DA | DA |
| Noah_D | 0.15 | 0.12 | 0.22 | 0.20 |
| CLM_D | 0.16 | 0.12 | 0.20 | 0.18 |
| SMAP L4C gross primary productivity ($g\ m^{-2}\ d^{-1}$) | 7.30 | | 8.10 | |
| Modeled gross primary productivity ($g\ m^{-2}\ d^{-1}$) | No DA | DA | No DA | DA |
| Noah_D | 4.70 | 3.83 | 7.42 | 5.45 |
| CLM_D | 5.84 | 5.88 | 10.10 | 4.51 |
| CASTNET (MLM-calculated) daytime $V_{d[ozone]}$ ($cm\ s^{-1}$) | 0.39 | | 0.39 | |
| Modeled daytime $V_{d[ozone]}$ ($cm\ s^{-1}$) | No DA | DA | No DA | DA |
| Noah_D | 0.70 | 0.66 | 0.89 | 0.68 |
| CLM_D | 0.74 | 0.75 | 1.07 | 0.52 |
| Noah_W | 0.63 | 0.62 | 0.82 | 0.79 |
| CASTNET daytime $F_{t[ozone]}$ ($nmol\ m^{-2}\ s^{-1}$) | 8.67 | | 11.66 | |
| Modeled daytime $F_{t[ozone]}$ ($nmol\ m^{-2}\ s^{-1}$) | No DA | DA | No DA | DA |
| Noah_D | 17.95 | 17.15 | 30.59 | 24.27 |
| CLM_D | 18.85 | 18.93 | 35.63 | 19.24 |
| Noah_W | 16.90 | 16.48 | 29.44 | 28.09 |



**Figures**

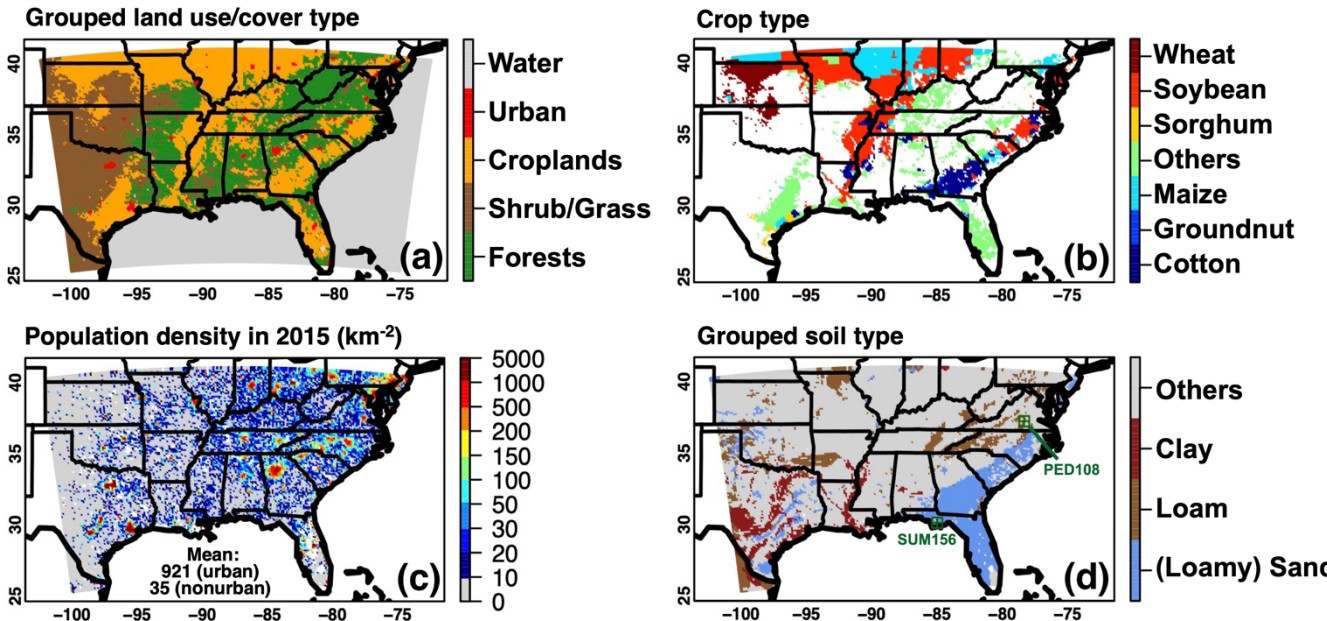

Figure 1: (a) Grid-dominant land use/land cover types grouped from the original 20-category model input (Figure 1c
in Huang et al., 2021) based on the method in Table S1; (b) grid-dominant crop type over cropland-dominant
regions; (c) gridded population density in 2015; (d) highlighted grid-dominant soil types of sand/loamy sand, loam,
and clay which are most relevant to discussions in this paper. The original soil type input from the State Soil
Geographic database is shown in Figure S1 in Huang et al. (2021). Locations of the two CASTNET sites for the case
studies are denoted in green.

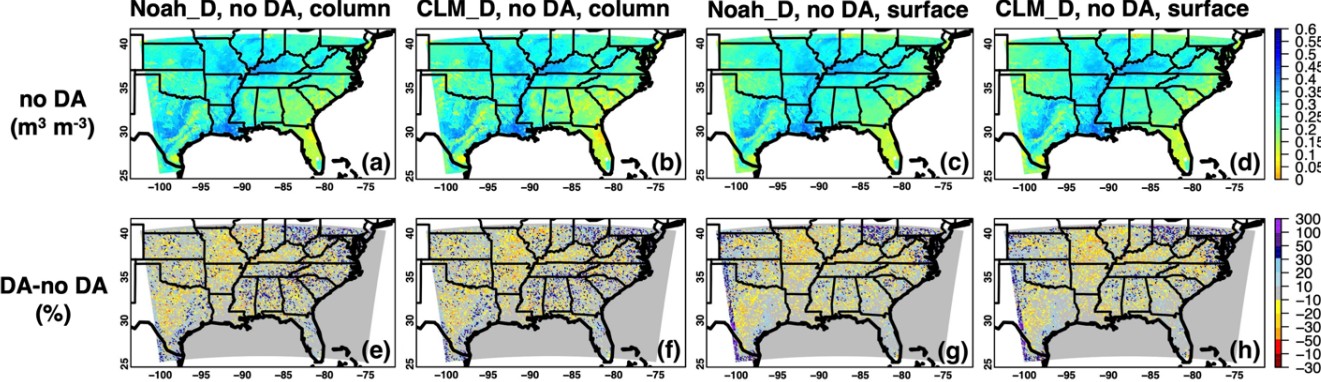

Figure 2: Period-mean (16–28 August 2016) WRF-Chem (a-b) column-averaged and (c-d) surface-layer soil moisture
fields at initial times and (e-h) their relative changes in % due to the SMAP DA. Results based on the Noah_D and
CLM_D cases are shown in (a;c;e;g) and (b;d;f;h), respectively.



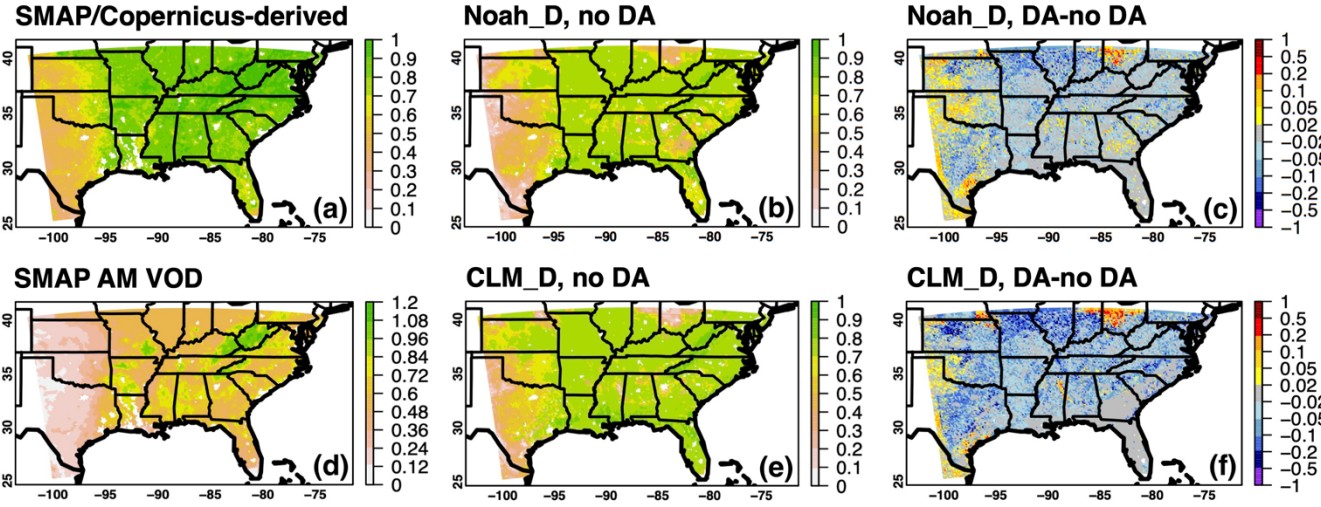

**Figure 3: Period-mean (16–28 August 2016) green vegetation fraction (a) derived from the Copernicus Global Land Service product and the SMAP morning-time (AM) vegetation optical depth (VOD) using the method described in**
**Figure S2; (b;c;e;f) based on WRF-Chem calculations as well as their responses to the SMAP DA. Results from the Noah_D and CLM_D cases are shown in (b-c) and (e-f), respectively. Period-mean SMAP AM VOD is shown in (d).**

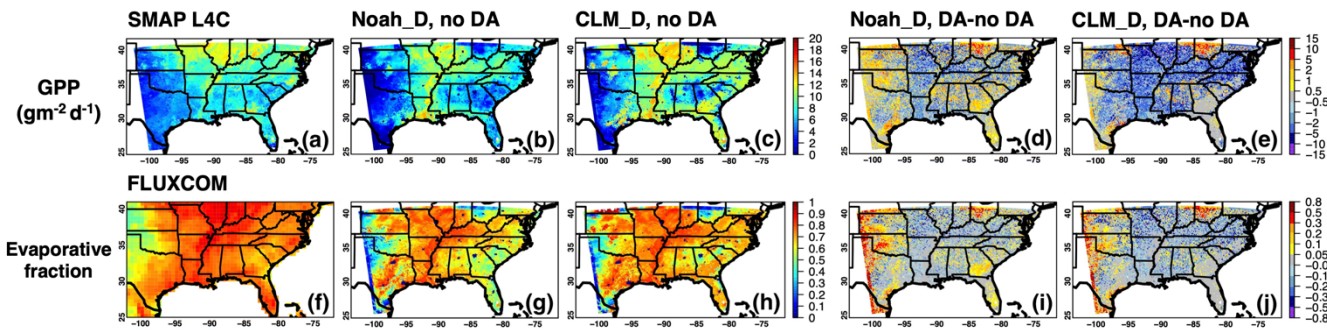

**Figure 4: Period-mean (16–28 August 2016) WRF-Chem calculated (b-e) gross primary productivity (GPP); and (g-j)**
**evaporative fraction as well as their responses to the SMAP DA. Results based on the Noah_D and CLM_D cases are shown in (b;g;d;i) and (c;h;e;j), respectively. Period-mean SMAP L4C GPP and FLUXCOM evaporative fraction are shown in (a;f), which are also used to evaluate the model results (Table 3).**

**Figure 5:** Period-mean (16–28 August 2016) WRF-Chem calculated daytime (a-c) surface air temperature and (g-i) surface radiation as well as (d-f;j-l) their responses to the SMAP DA. Results based on the Noah_D, CLM_D, and P1_W cases are shown in (a;d;g;j), (b;e;h;k), and (c;f;i;l), respectively. Overall, the weather fields from Noah_D and Noah_W (not shown in figures) cases are nearly the same. Grey lines in (a) indicate the B-200 flight paths over the southeastern US during the 2016 ACT-America campaign.






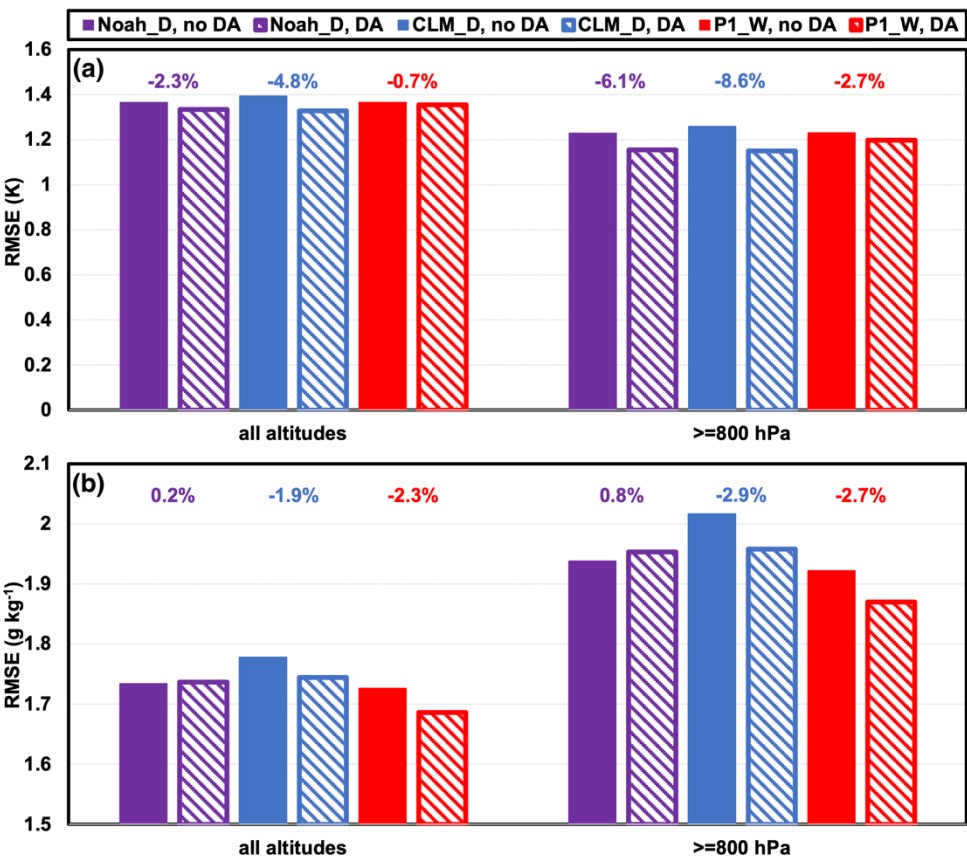

**Figure 6: Evaluation of (a) air temperature and (b) water vapor mixing ratios from several WRF-Chem simulations with the B-200 aircraft observations during the 2016 ACT-America campaign. The RMSEs are summarized in barplots based on model comparisons against observations at all altitudes and near the surface (≥800 hPa). Colored texts above the barplots indicate the SMAP DA impacts on RMSEs. The B-200 flight paths are indicated in Figure 5a.**





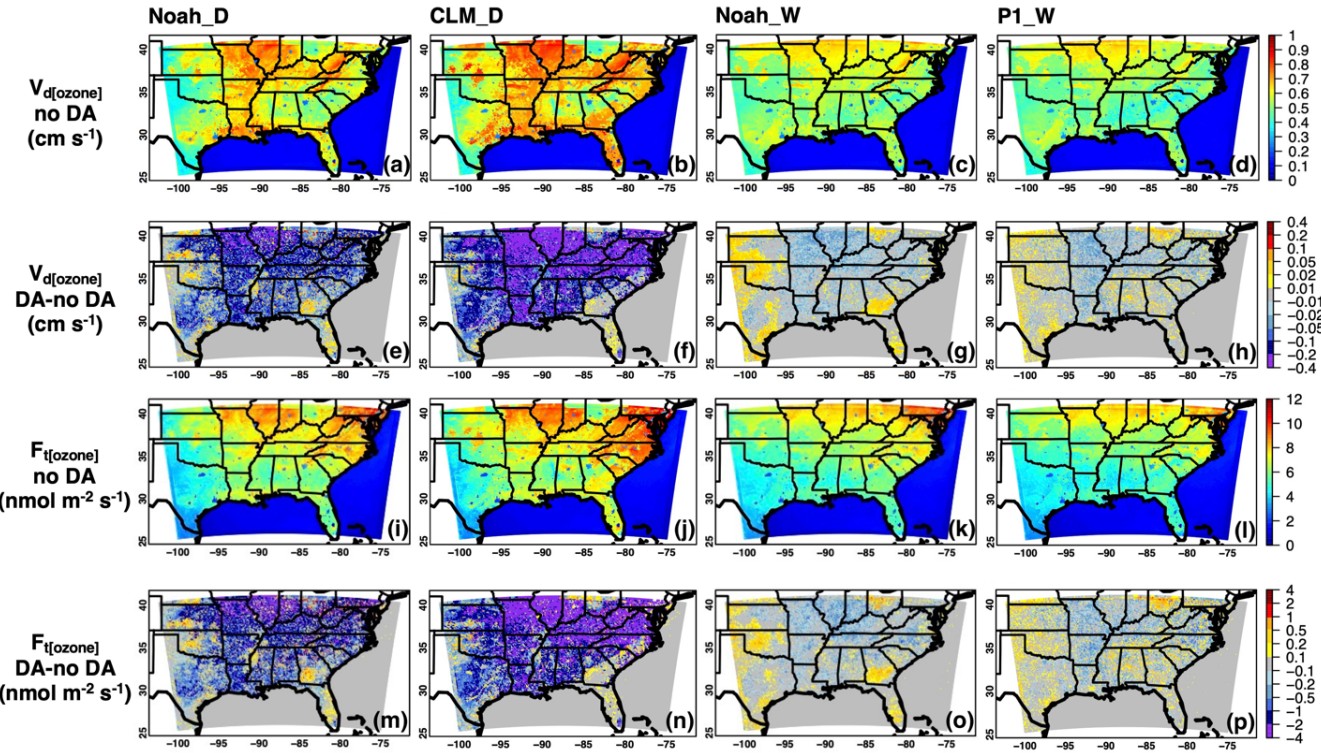

**Figure 7: Period-mean (16–28 August 2016) WRF-Chem (a-d) O₃ dry deposition velocity and (i-l) O₃ dry deposition flux, as well as (e-h; m-p) the impacts of SMAP DA on these model fields. Results are shown for (a;e;i;m) Noah_D, (b;f;j;n) CLM_D, (c;g;k;o) Noah_W, and (d;h;l;p) P1_W cases, averaged throughout the day.**



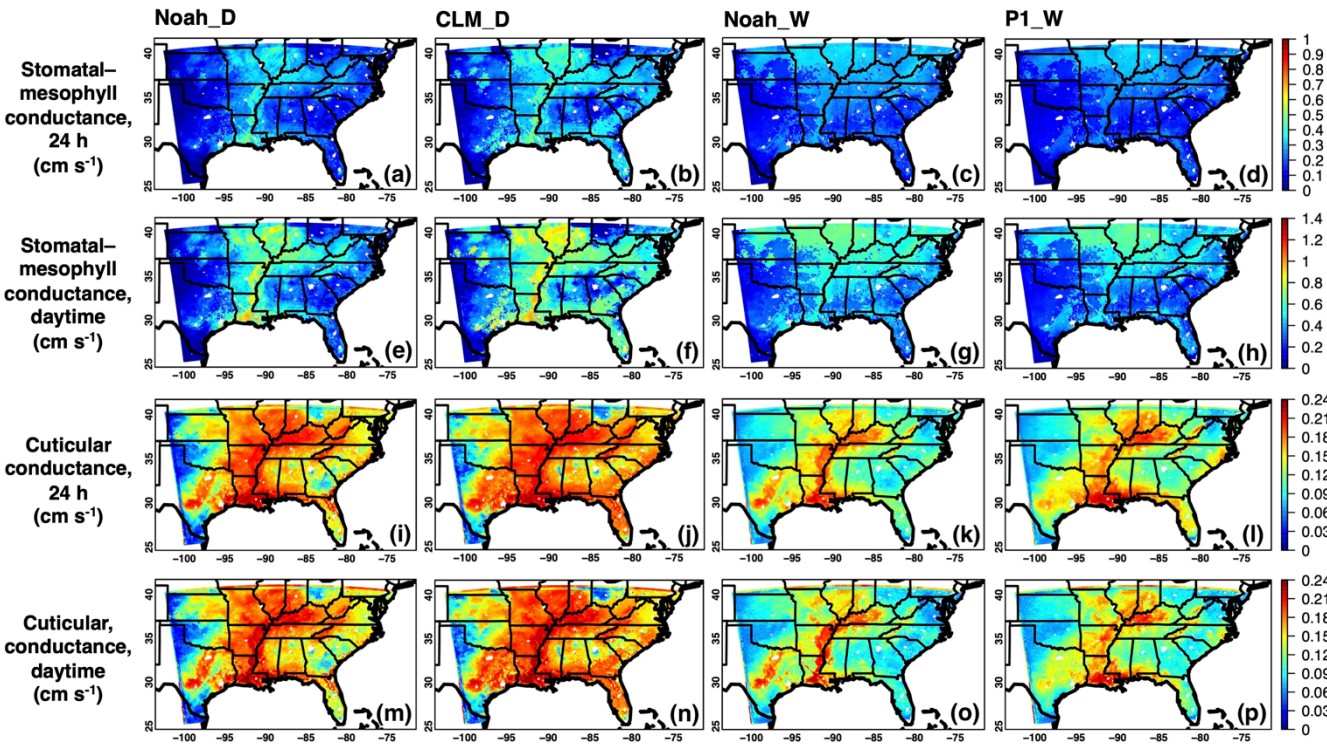

**Figure 8: Period-mean (16–28 August 2016) WRF-Chem (a-h) stomatal-mesophyll and (i-p) cuticular conductances over non-urban terrestrial regions. Results are shown for (a;e;i;m) Noah_D, (b;f;j;n) CLM_D, (c;g;k;o) Noah_W, and (d;h;l;p) P1_W no-DA cases, averaged (a-d;i-l) throughout the day and (e-h;m-p) during the daytime.**

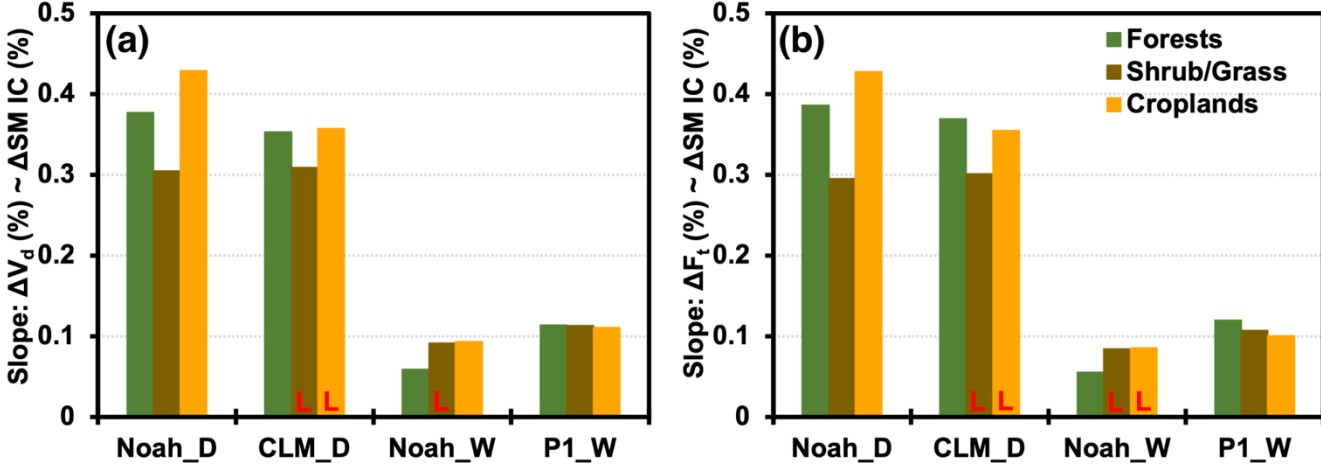

**Figure 9: Regression slopes of relative changes of (a) O₃ dry deposition velocity and (b) O₃ dry deposition flux versus the relative changes of column-averaged soil moisture initial conditions due to the SMAP DA, summarized by three LULC groups for all model cases listed in Table 1. The red "L"s indicate the regression analyses with low correlation (i.e., $r$ values of 0.39-0.49). The $p$ values for all regression analyses are <<0.01.**





985



**Figure 10: Period-mean (16–28 August 2016) diurnal cycles of (a-b) O₃ dry deposition velocity and (c-d) O₃ dry deposition flux based on the CASTNET dataset (in black solid lines) and their WRF-Chem counterparts (in purple, blue and brown lines) at the (a;c) SUM156 and (b;d) PED108 sites whose locations are shown in Figure 1d. (e-f) indicate the diurnal variability of WRF-Chem column-averaged soil moisture (normalized) at these two sites, and 990 grey vertical lines denote the initial times of WRF-Chem. Results from the no-DA and DA cases are indicated in solid and dashed lines, respectively.**





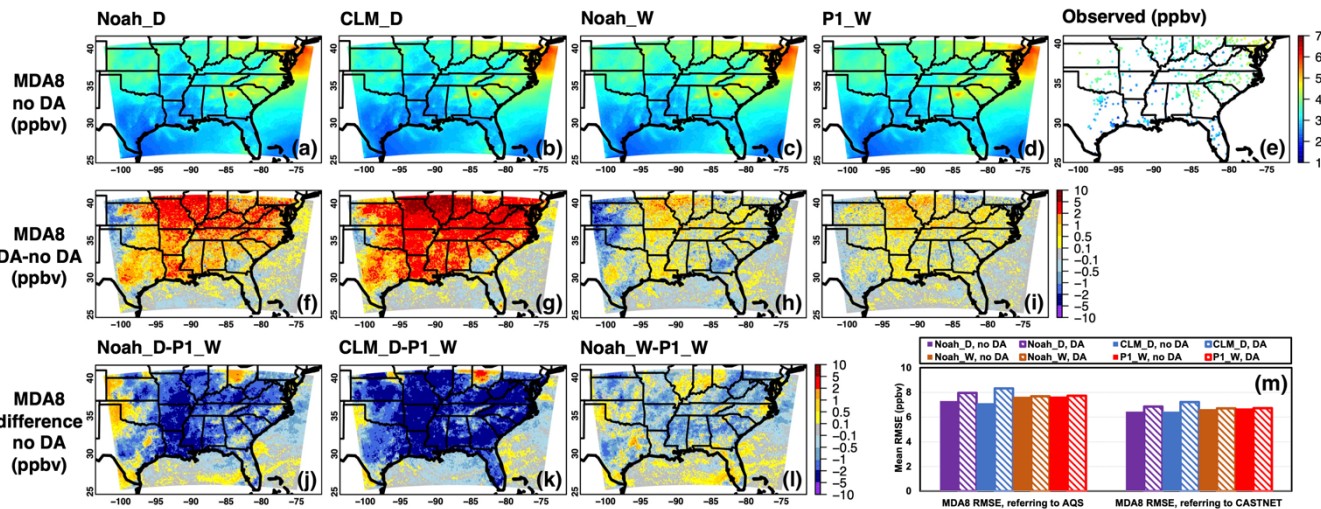

**Figure 11: Period-mean (16–28 August 2016) (e) observed and modeled (a-d) surface MDA8 O₃ fields and (f-i) their responses to the SMAP DA. Results based on the Noah_D, CLM_D, Noah_W, and P1_W cases are shown in (a;f), (b;g), (c;h) and (d;i), respectively, and the differences between the Noah-MP related cases and the P1_W case are shown in (j-l). Mean MDA8 RMSEs from various cases referring to surface observations are summarized in (m).**



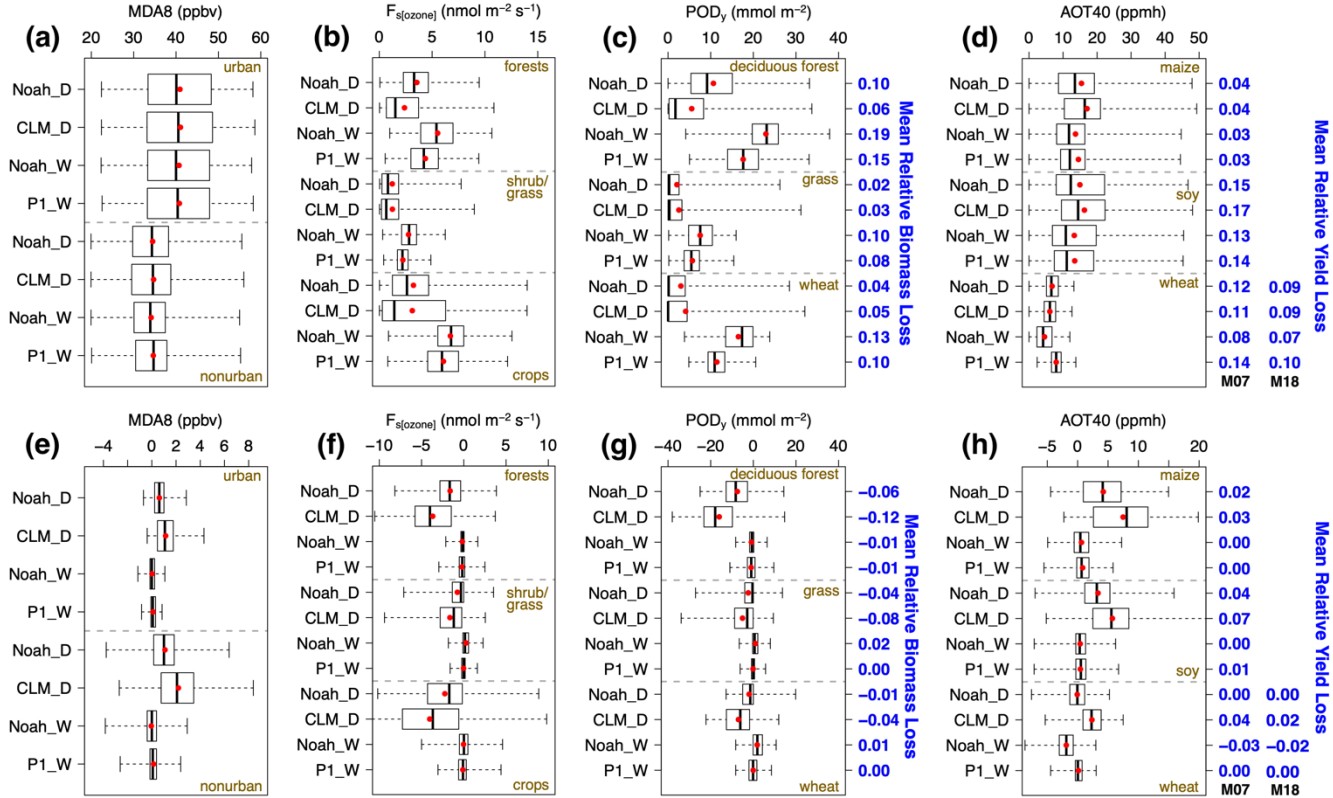

**Figure 12:** Box-and-whisker plots of WRF-Chem (a) MDA8 $O_3$; (b) daytime stomatal $O_3$ uptake $F_{s[ozone]}$; (c) derived $POD_y$; and (d) derived AOT40, summarized by LULC and crop types from all DA-enabled cases. The impacts of the SMAP DA on these model fields are shown in (e-h). Red filled circles indicate the mean values. The mean relative biomass/crop yield losses estimated based on all DA-enabled cases, as well as the SMAP DA impacts on these values, are included in (c-d;g-h) in blue text. The crop yield losses for wheat, estimated based on the derived AOT40 and two dose-response functions (M07: Mills et al., 2007; M18: Mills et al., 2018a) are included in (d;h).

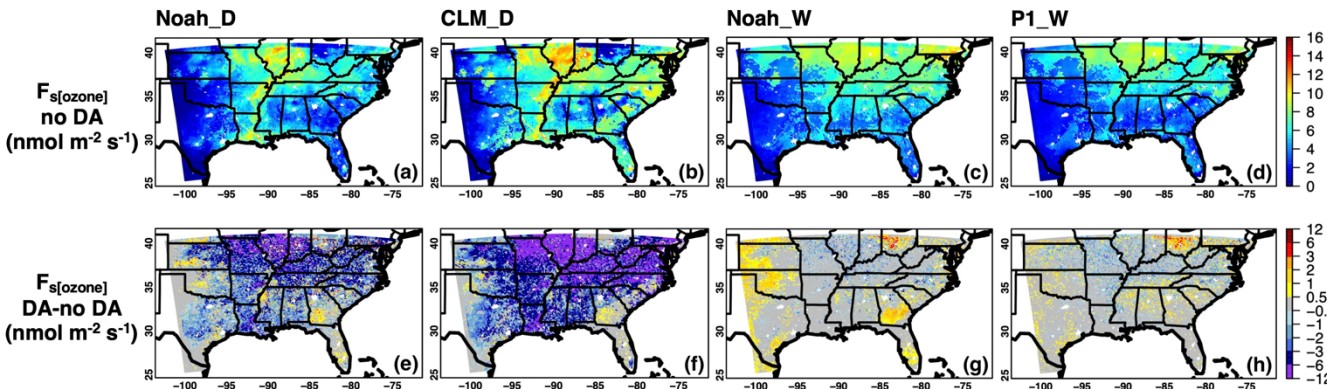

**Figure 13:** Period-mean (16–28 August 2016) WRF-Chem (a-d) daytime stomatal $O_3$ uptake $F_{s[ozone]}$ fields over non-urban terrestrial regions and (e-h) their responses to the SMAP DA. Results based on the (a;e) Noah_D, (b;f) CLM_D, (c;g) Noah_W, and (d;h) P1_W cases are shown.





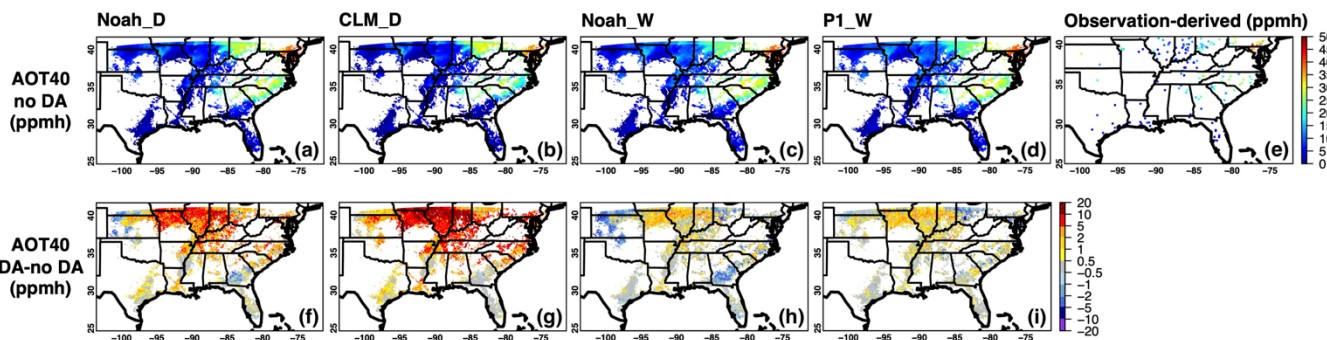

1015

**Figure 14: AOT40 in cropland-dominant model grids, derived from surface O₃ fields during 16–28 August 2016: (e) indicates the observation-derived, and those based on WRF-Chem results as well as their responses to the SMAP DA from the Noah_D, CLM_D, Noah_W, and P1_W cases are shown in (a;f), (b;g), (c;h), and (d;i), respectively.**

1020