# Peer review of "Satellite soil moisture data assimilation impacts on modeling weather variables and ozone in the southeastern US - Part 2: Sensitivity to dry deposition parameterizations"

_Atmospheric Chemistry and Physics, 2021_

## Referee Comment (RC1)

**Review ACP-2021-1068**

Satellite soil moisture data assimilation impacts on modeling whether variables and ozone in the southeastern US - Part 2: Sensitivity to dry deposition parameters

**1 general comments**

`evaluating the overall quality of the discussion paper`
The manuscript presents a comprehensive comparison of various vegetation relevant variables (e.g. soil moisture, GPP, surface air temperature) modeled with and without soil moisture assimilation. The main feat is the implementation of a data assimilation of soil moisture in two widely used land surface models NOAH-MP and CLM within WRF-chem. The manuscript continues to explore the effect of assimilated soil moisture on the ozone dry deposition. The authors compare the dynamic dry deposition schemes of the NOAH-MP and CLM (Bell-Berry type stomatal resistance) with the Wesley scheme of NOAH-MP (Javis-type stomatal resistance) and NOAH (Javis-type stomatal resistance). Ultimately, they extrapolate their resulting ozone surface concentrations from 2 weeks (Aug 16–28) to vegetation ozone damage risk indices MDA8 and AOT40. All studies are validated against observational data.

- The manuscript is overall well written and

- addresses globally relevant issues.

- It remains, however, unclear what distinguish the *Wesely* and *dynamic* schemes (see specific comment)

- Some remarks need citations (see specific comments)

- Some figures / captions are hard to understand and might need a better explanation in the text

**2 specific comments**

`individual scientific questions/issues`

- P1L14: *"Realistically representing this process in models is important for accurately simulating* $O_3$ *concentrations and exceedances [...]"* In general, I agree with this opening statement. Though, the method of assimilating soil moisture observation into the model does make the results partly more realistic, no new approaches to ozone dry deposition different from the usual resistance analogous one are explored. How realistic this is from a micro-physical / micro-meteorological perspective is disputable.
  The authors may change the sentence slightly: "The representation of this process in models is [...]"

- Section 2
  - The authors discriminate "Wesely" and "dynamic" scheme as well as "Ball-Berry" and "Javis" stomatal resistance. In general, these only differ by including a coupling to the leaf area index (LAI) via a soil moisture and temperature dependent photosynthesis ($V_{\max}$) which is also referred to as $\beta$ schemes. The dynamic scheme also takes the canopy density (shaded vs sun-lit leaves) into account. This could be made clearer in the beginning of the section. A coupling to soil moisture and dynamic LAI could be, in principle, also achieved with a Javis-type stomatal resistance model (see *ICP Mapping Manual - Chapter III: Mapping Critical Levels for Vegetation* – Mills et al. (2017)).
  - P4L126: The authors should include (and ideally list) the model resolutions used for this study as well.
  - P5L163: Is there a reason to distinguish small and capital letter stomatal resistances? Do they refer to leaf and canopy level or bulk resistances? The manuscript doesn't say so. Please indicate the meaning of this notation.
  - P6L179: $V_{\max}$ vs $V_{\mathrm{d}}$ (P2L47). It is slightly unfortunate to use capital V for both dry deposition velocities and maximum carboxylation rate. The authors may consider using $v_{\mathrm{d}}$ for dry deposition velocities.
  - P5L216: Does the equation here refer to the Javis-type stomatal resistance used in the NOAH version?
  - P5L216: "$\sim 9999, > 40^{\circ}\mathrm{C}\,\mathrm{or}\,T_s < 0^{\circ}\mathrm{C}$" is not a good notation. Rather write "else". 9999 appears to be an arbitrarily large number and should be commented on.
  - P6L184–184: It is not clear, how (and if so) the $\beta$ schemes of CLM and NOAH-MP differ in their mathematical formulation.
  - P6L184: Here and probably also in other occasions, it should be made clear which CLM version and configuration the authors compare to.
  - P6L191: Regarding the Monin-Obukhov similarity theory, different formulations for the universal functions exist. It is known, that the MO similarity theory is not applicable at turbulence resolving resolutions (e.g. large Eddy simulations) and is also challenged by mountainous terrain. This should be taken into consideration at this point. For a discussion see, e.g. Basu and Lacser (2017) and Emeis et al. (2018).
  - P8/9L252–L271: Including a comprehensive table summarizing the key information on the datasets used in Part I and II of this study would be beneficial. Key information should include resolution, temporal extend, observed variables, etc.
  - P9L274: "*MDA8* $\mathrm{O}_3$ *fields over unban and nonurban regions were investigated [...]*" How do the authors account for and assess the effective ozone titration

in the proximity of urban conglomerates? Please elaborate on the skill of WRF-chem in this regard.

– P9L281-L282: *"Our 13-days WRF-chem model results were linearly-extra-polated to ~three months to derive the* $POD_y$ *and AOT40 fields."* For a mere comparison concerning the effect of soil moisture assimilation and dry deposition schemes, this extrapolation may not be necessary. This extrapolation from 2 weeks in August to three months appear not to be robust especially in a dynamic scheme. The authors may consider dropping parts of section 3.3 and focus on the core study of their manuscript.

• Section 3

– P10L313/L215: *"hydrological regime"* and *slightly drier* Has CLM been run with hydraulic stress module (Kennedy et al., 2019)? This would affect LAI and stomatal resistance.

– P11L223/334: *"Referring to the satellite-derived GVF fields which are also subject to large uncertainty, [...]"* How large is "large" with respect to the magnitude of the differences between models and schemes discussed?

– P12L353/354: *"[...] been reported in previous studies."* Can you cite these studies?

– P17L519: How has $F_s$ been estimated or is it the same as $F_t$? It is not clear from the manuscript whether this is a direct output from the model or has been obtained from other output variables, e.g. $F_t$ which was derived from $V_d$.

– P13L400 and Fig.7: *"Figure 7 presents the period-mean, daily averaged* $V_d$ *and dry deposition flux* $F_t$ *[...]"* Taking the results presented in Figure 10 into consideration, does it make sense to present $V_d$ and $F_t$ as daily averages? Effectively, all model integrations show very similar nighttime dry deposition. It could be more informative to show and discuss their day time or noon $(12 \pm 2\,\mathrm{h})$ averages.

– P13L405: *"[...] many existing model- and measurement-based studies [...]"* Citations?

– P14L422–423: Would it be possible to indicate the $r$ values in Figure 9? How large are the uncertainties associated with the slopes?

– P15L558–460: Regarding the bias of the MLM derived dry deposition velocities as mentioned on P9L67, would it be possible to correct for this? Or at least make a more comprehensive statement about the nature of this bias as it affects your model performance evaluation.

– P15L464: *"responded least significantly"* Consider rephrasing as no statistical test seems to back up the use of the term "significant". Regarding Figure 10, there are no standard deviations or other indicators of uncertainty or variability displayed to help assess the improvement of model skill. If possible

please include such. It would be, in general, interesting to have a look at the relative change of different contributions to $V_d$, e.g. $R_s$, $R_a$. Such assessment has already been suggested by Hardacre et al. (2015).

– P17L539–541: *"Based on the known seasonal variability of surface* $O_3$ *and* $V_d$ *in the study region, the linearly scaled* $POD_y$ *and AOT40 values may have been overall underestimated."* This is a strong statement, and you should gve some references regarding "the known seasonal variability". E.g. Fig. 2 in Strode at al. (2015) does not quite support this conclusion for all regions of the US. The RMSE in Figure 11m points to a difference between observation and modelled MDA8 of the order of 10–16 %! Which you confirm to be "positively biased" (P16L503). This in conjunction with the very short model integration of about 2 weeks does not support such a strong claim of "underestimation". You should rephrase this conclusion or make a more robust assessment.

- Section 4 should more clearly address the order of magnitude of the different uncertainties (model and observation) relevant to the presented analyses.

- Figure 11 is quite busy. You may consider moving panel m to a Figure of its own.

- Figure 10, the difference between observation and model results should be more thoroughly discussed. E.g. where does the large difference in the temporal extent and timing of $V_d$ come from?

**3 technical corrections**

purely technical corrections

- P1L18: I'm missing the acronym WRF-chem.

- P5L136/L142/L148: *"lass access"* Typo "last"?

- P5L157: $GVF$ should read GVF.

- P7L204/L209: The "[ ]" are not necessary.

- P7L211: *")surface"* missing whitespace.

- P9L274: *unban* Typo "urban"

- P9L282 and others: $\sim$*three months* use of $\sim$ in text body should be avoided. Maybe use "approximately" or "roughly" or "about" instead.

- P15L554: *"[...] dramatically higher [...]"* The authors may consider using a more neutral formulation.

- P16L515: *"aggressive efforts"* The authors may consider using a more neutral formulation.

- Figure 3/4: '-' should probably read '$-$'?

---

## Author Comment (AC1)

**Author response to reviews**

The authors appreciate ACP and the referees' efforts and constructive comments. Please see below a list of major changes followed by our point-by-point response (in blue) to all general and specific comments (in black). Quoted text from the revised manuscript is *in italic*.

Major changes have been made to the manuscript to address both referees' comments, including:

- Added a table (current Table 2) that summarizes all relevant evaluation datasets along with their attributes.
- Clarified, updated, and/or discussed the methods regarding the two dry deposition schemes (i.e., the one-big-leaf multiplicative Wesely scheme versus the two-big-leaf photosynthesis-coupled) and their differences, the Noah and CLM types of $\beta$ functions, the model representations of SM-vegetation growth relationship, and $F_s$ calculations following the method in CLRTAP (2017). These include adding or updating relevant equations in the manuscript and the SI.
- Introduced and discussed the uncertainty and limitations associated with the satellite-derived vegetation data and the CASTNET $v_d$ product based on the MLM calculations.
- Extended the analyses and discussions on the temporal and spatial representativeness of our results as well as the scale dependencies of $v_d$ and $O_3$ modeling. These include, on the temporal scale, diurnal-to-daily variability of $O_3$ dry deposition fluxes and their major pathways, as well as 13-day versus seasonal $O_3$ metrics in 2016 and other years; and on the spatial scale, representation errors in the comparisons of point- and regional-scale model fluxes due to surface heterogeneity within the model grids and the pixels of the assimilated satellite SM data, as well as the likely unrealistically-represented vertical mixing in regional models and the limitations in the Monin-Obukhov similarity theory (MOST).
- Extensively edited figures and their captions, as well as improved referencing and citations in all sections of the manuscript and the SI.

**Response to RC1**

general comments - evaluating the overall quality of the discussion paper
The manuscript presents a comprehensive comparison of various vegetation relevant variables (e.g. soil moisture, GPP, surface air temperature) modeled with and without soil moisture assimilation. The main feat is the implementation of a data assimilation of soil moisture in two widely used land surface models NOAH-MP and CLM within WRF-chem. The manuscript continues to explore the effect of assimilated soil moisture on the ozone dry deposition. The authors compare the dynamic dry deposition schemes of the NOAH-MP and CLM (Bell-Berry type stomatal resistance) with the Wesley scheme of NOAH-MP (Javis-type stomatal resistance) and NOAH (Javis-type stomatal resistance). Ultimately, they extrapolate their resulting ozone surface concentrations from 2 weeks (Aug 16–28) to vegetation ozone damage risk indices MDA8 and AOT40. All studies are validated against observational data.

The manuscript is overall well written and addresses globally relevant issues. It remains, however, unclear what distinguish the Wesely and dynamic schemes (see specific comment). Some remarks need citations (see specific comments). Some figures/captions are hard to understand and might need a better explanation in the text.
Thank you for the overall positive feedback and useful suggestions. As noted in the list of major changes above, we clarified the differences between the two dry deposition schemes (i.e., the onebig-leaf multiplicative Wesely scheme versus the two-big-leaf photosynthesis-coupled) in the revision. Figures and their captions have been extensively edited and supported by discussions in the text. Referencing and citations have been improved accounting for all specific comments.

specific comments - individual scientific questions/issues
P1L14: "Realistically representing this process in models is important for accurately simulating O3 concentrations and exceedances [...]" In general, I agree with this opening statement. Though, the method of assimilating soil moisture observation into the model does make the results partly more realistic, no new approaches to ozone dry deposition different from the usual resistance analogous one are explored. How realistic this is from a micro-physical / micro-meteorological perspective is disputable. The authors may change the sentence slightly: "The representation of this process in models is [...]"
The abstract has been revised substantially, now not including the word of "realistically". A change has been made to a sentence in Section 4 that previously contained "realistically". The manuscript now better describes the temporal and spatial representativeness of our results as well as the scale dependencies of $v_d$ and $O_3$ modeling, covering the limitations of the MOST as this referee pointed out.

Section 2 The authors discriminate "Wesely" and "dynamic" scheme as well as "Ball-Berry" and "Javis" stomatal resistance. In general, these only differ by including a coupling to the leaf area index (LAI) via a soil moisture and temperature dependent photosynthesis (Vmax) which is also referred to as β schemes. The dynamic scheme also takes the canopy density (shaded vs sun-lit leaves) into account. This could be made clearer in the beginning of the section. A coupling to soil moisture and dynamic LAI could be, in principle, also achieved with a Javis-type stomatal resistance model (see ICP Mapping Manual - Chapter III: Mapping Critical Levels for Vegetation – Mills et al. (2017)).
This is an important point. The structural differences between the Wesely and the "dynamic" dry deposition scheme (in chemistry routines) are now more clearly indicated in Sections 1 and 2.3. And note that in some of the revised dry deposition schemes, "*stomatal conductance is calculated based on the one big-leaf, multiplicative algorithms that are more complicated than the Wesely (1989) approach, in the way that the empirical maximum stomatal conductance is adjusted by more factors, including water availability and vegetation attributes*". The differences between "Ball-Berry" and the multiplicative "Jarvis" stomatal resistance options in the Noah-MP land surface model (LSM) are discussed in Section 2.2. The empirical maximum stomatal conductance used in multiplicative algorithms such as Wesely and Jarvis can introduce uncertainty. We also extended the introduction to the β schemes in the Noah-MP LSM in Section 2.2, taking this referee's later comments.

P4L126: The authors should include (and ideally list) the model resolutions used for this study as well.
Yes, it is necessary to state that the previous 12 km/63 vertical layer grid was implemented in this study. The 25 km grid in Huang et al. (2021) was not used in this work.

P5L163: Is there a reason to distinguish small and capital letter stomatal resistances? Do they refer to leaf and canopy level or bulk resistances? The manuscript doesn't say so. Please indicate the meaning of this notation. P6L179:  Vmax vs Vd (P2L47). It is slightly unfortunate to use capital

V for both dry deposition velocities and maximum carboxylation rate. The authors may consider using vd for dry deposition velocities.

Thanks for the two sets of suggestions above. The capitalizations of resistances and deposition velocity have been updated and their current forms are more consistent with the general usages in literature. Only $r_{s,i}$ (where i = sunlit or shaded, as noted in text) stands for the leaf-level variables, all others are for bulk.

P5L216: Does the equation here refer to the Javis-type stomatal resistance used in the NOAH version?

The Wesely dry deposition scheme introduced here was used in the cases that implemented Noah (part 1) and Noah-MP (new in this work) LSMs, with Noah-type β. Please see Table 1 for details.

P5L216: "~9999, >40 °C or Ts <0 °C" is not a good notation. Rather write "else". 9999 appears to be an arbitrarily large number and should be commented on.

This equation is written based on equations (3) and (4) of Wesely (1989) and the implementation in WRF-Chem. The WRF-Chem implementation of 9999 is consistent with the note in Wesely (1989) that outside this range (i.e., $T_s$ >40 °C or <0 °C), stomatal resistance is set to a very large value, to implement the assumption that the transfer through stomata is stopped. A clarification has been added here.

P6L184–184: It is not clear, how (and if so) the β schemes of CLM and NOAH-MP differ in their mathematical formulation.

The mathematical formulations of the CLM and Noah types of β have been added here, according to equations (12) and (13) in Niu et al. (2011).

P6L184: Here and probably also in other occasions, it should be made clear which CLM version and configuration the authors compare to.

The CLM-type β and Ball-Berry stomatal resistance in Noah-MP are default in CLM version 4.5 and some of its earlier versions. This information has been added to the text and Table 1. Other LSM physics configurations can be found in Section 2.2.

P6L191: Regarding the Monin-Obukhov similarity theory, different formulations for the universal functions exist. It is known, that the MO similarity theory is not applicable at turbulence resolving resolutions (e.g. large Eddy simulations) and is also challenged by mountainous terrain. This should be taken into consideration at this point. For a discussion see, e.g. Basu and Lacser (2017) and Emeis et al. (2018).

Yes, there are different formulations. There are two $C_H$ schemes in Noah-MP. We clarify that the $C_H$ scheme used in this work is based on more general MOST, as defined in equation (16) in Niu et al. (2011). The main difference between this option and the previously used Chen97 option is that this scheme accounts for zero-displacement height while its does not distinguish the roughness lengths for heat and momentum. Some of the limitations of Chen97 that are related to the roughness lengths have been discussed in the SI of Huang et al. (2021).

We recognize the limitations of MOST which is used in many models/studies, and that it is difficult to realistically represent vertical mixing in regional-scale model simulations, as noted in numerous Large Eddy Simulation (LES) studies and pointed out by this referee. These can affect the

simulated $O_3$ distributions and processes, in part through some of the dry deposition pathways such as $r_a$ - as described in the case study (Section 3.2.2), MOST is used in Wesely and dynamic schemes whereas MLM uses a simpler approach based on wind fields, and this partially explains the differences between the WRF-Chem modeled and MLM $v_d$ values. The nighttime $v_d$ values from WRF-Chem and MLM, which contributed mostly by $r_a$, $r_b$ and non-stomatal $r_c$ pathways, were compared with flux observations at European forest sites during both dry and wet periods in the past decades (Lin et al., 2020) to help understand the magnitude of their associated uncertainty. We acknowledge that it would be more challenging to conduct modeling and measurement studies over mountainous terrains in general, as discussed in Emeis et al. (2018) and our previous studies. Note that complex terrains contribute to only a small portion of the study region as shown in Figure 1a of Huang et al. (2021), and elevations and terrains of the two CASTNET sites have been added to current Table 6. In Section 4, we cited the Makar et al. (2017) study that discusses representing canopy turbulences concerning the limitations of MOST and the potential use of LES for process-level study and improving parameterizations for canopy turbulence at regional chemical transport model resolutions.

P8/9L252–L271: Including a comprehensive table summarizing the key information on the datasets used in Part I and II of this study would be beneficial. Key information should include resolution, temporal extend, observed variables, etc.
A table (current Table 2) summarizing key information for all relevant datasets has been added.

P9L274: "MDA8 O3 fields over unban and nonurban regions were investigated [...]" How do the authors account for and assess the effective ozone titration in the proximity of urban conglomerates? Please elaborate on the skill of WRF-chem in this regard.
As introduced in this section, the model results are evaluated with observations, and, as shown in current Figures 11 and 12, the modeled urban-nonurban gradients are found in surface $O_3$ observations in/around many major cities such as Atlanta, Dallas, and multiple mid-Atlantic cities. We recognize that the model representation of $O_3$ titration over urban areas, which depends largely on its meteorological fields and emissions, affects the model performance of $O_3$ over these regions and their downwind areas. In general, this source of uncertainty has weaker impacts on model performance during the daytime of warm seasons than nighttime/cold seasons. Also, based on our experiences in multiscale $O_3$ modeling covering other urban areas (e.g., Figure 3f in Huang et al., 2011, https://doi.org/10.5194/acp-11-3173-2011), we believe that this process is better modeled by WRF-Chem at 12 km resolution than in coarse-resolution model simulations in which emissions are more diluted, surface heterogeneity and fine-scale meteorological processes (e.g., sea-land breezes for urban-water interfaces) are poorly represented.

We want to also point out that, Huang et al. (2021) covers the evaluation of the used anthropogenic emission inventory and the impact of SM DA on titration: "Faster winds and thickened PBL dilute air pollutants including $O_3$ and its precursors and therefore reduce $O_3$ destruction via titration (i.e., $O_3+NO \rightarrow O_2+NO_2$) as well as photochemical production of $O_3$". In Section 3.3.1 of this work we also note the urban-nonurban interactions contributing to the MDA8 responses to the DA: "*the DA impacts on MDA8 reach 3–4 ppbv in places, under the controls of the local-to-regional circulation patterns (Figure 13a, e)*".

P9L281-L282: "Our 13-days WRF-chem model results were linearly-extrapolated to ~three months to derive the PODy and AOT40 fields." For a mere comparison concerning the effect of soil moisture assimilation and dry deposition schemes, this extrapolation may not be necessary. This extrapolation from 2 weeks in August to three months appear not to be robust especially in a dynamic scheme. The authors may consider dropping parts of section 3.3 and focus on the core study of their manuscript.

*While we focus on "qualitatively interpreting the results and discussing their implications" here, we agree that additional discussions/justifications should be added on relating the 13-day results to seasonal conditions. We now provide additional information in the SI and text on seasonal surface $O_3$ concentrations and surface fluxes, to support the discussions in Section 3.3. Please see an extended reply to your comment on L539-541.*

Section 3
P10L313/L215: "hydrological regime" and slightly drier Has CLM been run with hydraulic stress module (Kennedy et al., 2019)? This would affect LAI and stomatal resistance.

*The β scheme controls the water stress factor in model. The CLM type of β scheme in Noah-MP has been used in CLM version 4.5 and some of its earlier versions. This is discussed in Kennedy et al. (2019), in comparison with a leaf water potential based water stress factor that became available in CLM version 5. Indeed, a key finding of this study is that the representations of hydraulic stress affect various model fields in both the no-DA and DA cases.*

P11L223/334: "Referring to the satellite-derived GVF fields which are also subject to large uncertainty, [...]" How large is "large" with respect to the magnitude of the differences between models and schemes discussed?

*Uncertainty discussions have been included in Figure S2 caption, and referred to in text: "The accuracy of the satellite-derived GVF fields can be affected by: 1) the quality of the original Copernicus GVF product, which has an overall slight positive bias of 0.02 (4.0%) relative to ground-based observations, and such biases are land cover dependent (Copernicus Global Land Operations, 2020); 2) the uncertainty in the original SMAP VOD retrievals, which may be reduced or canceled as the ratios of period-mean/climatological VOD were applied in the calculation; 3) the temporal representativeness of the 10-day average Copernicus GVF product as the land surface conditions under cloudy and poor atmospheric conditions cannot be sampled; and 4) this approach used to derive the period-mean GVF and the assumptions associated with it. In the discussions in Section 3.1, we assume that 1) is the main source of uncertainty of these satellite-derived GVF fields, and according to Copernicus Global Land Operations (2020), positive biases are very likely to be associated with the GVF data exceeding 0.6 over forests and croplands and those falling within 0.2−0.6 over grasslands".*

P12L353/354: "[...] been reported in previous studies." Can you cite these studies?
*This point has been discussed in our part 1 study which is now cited here.*

P17L519: How has Fs been estimated or is it the same as Ft? It is not clear from the manuscript whether this is a direct output from the model or has been obtained from other output variables, e.g. Ft which was derived from Vd.

*The stomatal flux $F_s$ is the portion of $F_t$ that enters the plants' stomata and therefore is different from $F_t$. This is noted in equation (17) as "stomatal uptake". It was earlier calculated from stomatal*

conductance and $O_3$. We now follow the approach specified in CLRTAP (2017) to calculate $F_s$, which is based on the following equation (current equation 19):

$$F_s = C \ (nmol \ m^{-3}) \ \times \ g_s \ \times \ \frac{r_c}{1.3 \times 150 \times \sqrt{\frac{L}{u}} + r_c}$$

where $g_s$, $L$, and $u$ are stomatal conductance, leaf width (0.04 m in Noah-MP) and wind speed, respectively. Based on this equation, the $F_s$ calculations have been redone, leading to the updates to Figures 12 and 13 (current Figures 13 and 14).

P13L400 and Fig.7: "Figure 7 presents the period-mean, daily averaged Vd and dry deposition flux Ft [...]" Taking the results presented in Figure 10 into consideration, does it make sense to present Vd and Ft as daily averages? Effectively, all model integrations show very similar nighttime dry deposition. It could be more informative to show and discuss their day time or noon (12 ± 2 h) averages.

Daily $v_d$ and $F_t$ are shown in Figure 7 because in discussions they are often related to 24 h-averaged FLUXCOM and GPP/SIF data in terms of their spatial variability and the level of confidence in model performance. The diurnal variability of the major terms of $v_d$, are indicated in Figures 8 and 10, Table 5 and Section 3.2.2 text. As shown below, the daytime averaged $v_{d[ozone]}$ (cm s$^{-1}$) values in the upper panels display similar spatial patterns as the daily averaged $v_{d[ozone]}$ (cm s$^{-1}$) in the lower panels. The fluxes around midday are discussed in text referring to Figures 10 and S5: "*The slight declines in fluxes around midday based on some simulations can result from the water and heat stresses which cause stomata closures (Figure 10c, d). The water stress starts to get relieved since the mid-afternoon at the SUM156 site under the influences of convective precipitation whereas persists throughout the afternoon at the PED108 site (Figure 10g, h), which helps shape the slightly different afternoon flux dynamics at these two locations*". Based on this we do not think noon-time flux averages are most representative of the daytime conditions.

[Figure]

P13L405: "[...] many existing model- and measurement-based studies [...]" Citations?
A few citations have been added: "*(e.g., Val Martin et al., 2014; Hardacre et al., 2015; Silva and Heald, 2018; Lin et al., 2019)*".

P14L422–423: Would it be possible to indicate the *r* values in Figure 9? How large are the uncertainties associated with the slopes?
Two panels showing the *r* values and standard error (%) of the slopes from the regression have been added to Figure 9. The $F_t$ related results previously shown in panel (b) are now only introduced in the figure caption to address another comment.

P15L558–460: Regarding the bias of the MLM derived dry deposition velocities as mentioned on P9L67, would it be possible to correct for this? Or at least make a more comprehensive statement about the nature of this bias as it affects your model performance evaluation.

The operational MLM product is from the CASTNET database, and as noted, it may be highly uncertain. We already introduced that the MLM $v_d$ values are based on simplified approaches of calculating $r_a$ and $r_b$ as well as the empirical approach of calculating $r_s$ which have known limitations. New analysis has been conducted for this case study. For both sites, we note that many/most/all MLM assumptions apply (according to CASTNET site information table), and we show that the MLM $v_d$ data have very little daily variability during the study period (current Figure S5). It is likely that many but not all of these are filled historical average $v_d$ values due to the lack of meteorological measurements needed for the MLM calculations. The factual data such as plant and canopy attributes used as MLM inputs are outdated (i.e., 15+ year old, according to the CASTNET database). Additionally, representation errors are estimated to be pronounced when comparing the point-scale MLM fluxes with our 12 km WRF-Chem results due to the surface heterogeneity within our model grid cells and the satellite pixels (as shown in the Google Earth maps below, whose sizes are close to 10 km). Because of all these, no correction has been applied to the operational MLM data but the MLM related discussions have been extended in this section.

[Figure]

[Figure]

P15L464: "responded least significantly" Consider rephrasing as no statistical test seems to back up the use of the term "significant". Regarding Figure 10, there are no standard deviations or other indicators of uncertainty or variability displayed to help assess the improvement of model skill. If possible please include such. It would be, in general, interesting to have a look at the relative change of different contributions to Vd, e.g. Rs, Ra. Such assessment has already been suggested by Hardacre et al. (2015).

"Significantly" has been changed to "*strongly*". Standard deviation of $v_d$ and $F_t$ results are now included in Table 6. Figure 10 now contains additional panels showing the results of stomatal-mesophyll conductance $g_{sm}$. And now time series plots in current Figure S5 indicate the daily variability of $v_d$, $F_t$, $g_{sm}$ and SM anomaly. All of these added analyses support this statement.

P17L539–541: "Based on the known seasonal variability of surface O3 and Vd in the study region, the linearly scaled PODy and AOT40 values may have been overall underestimated." This is a strong statement, and you should give some references regarding "the known seasonal variability". E.g. Fig. 2 in Strode at al. (2015) does not quite support this conclusion for all regions of the US. The RMSE in Figure 11m points to a difference between observation and modelled MDA8 of the order of 10–16 %! Which you confirm to be "positively biased" (P16L503). This in conjunction with the very short model integration of about 2 weeks does not support such a strong claim of "underestimation". You should rephrase this conclusion or make a more robust assessment.

Section 4 should more clearly address the order of magnitude of the different uncertainties (model and observation) relevant to the presented analyses.

We now provide additional information in the SI and text on seasonal surface $O_3$ concentrations (based on AQS and CASTNET observations) and surface fluxes (inferred from carbon and energy fluxes according to the correlations between carbon/energy and $O_3$ fluxes which have been shown in earlier figures and discussions), to support the discussions in Section 3.3.

Based on the added analysis, the seasonal variability of surface $O_3$ concentrations over the SE US in 2016 is roughly similar to Figure 2c of Strode et al. (2015), while surface fluxes reached their peak values during June-July-August 2016. With the model biases being accounted for, the linearly scaled $POD_y$ and AOT40 values are overall underestimated referring to such seasonal variability of surface $O_3$ concentrations and fluxes. We also recognize the interannual variability in $O_3$ concentrations and fluxes that affects the impact assessments based on conditions of a particular year, and this is typical in studies that estimate $O_3$ impacts based on multiscale modeling results for a selected period (e.g., Lapina et al., 2014, Figure 3 and relevant discussions). We acknowledge that positive surface $O_3$ biases for this region is a common issue in global and regional models, and more efforts are in need to reduce them. Note that the RMSEs of the modeled $O_3$ from this work are close to, or much lower than the $O_3$ biases reported in Strode et al. (2015) and Lapina et al. (2014) for this region.

Figure 11 is quite busy. You may consider moving panel m to a Figure of its own.

Figure 11 has been reorganized. The previous Figure 11(e, m) are now Figure 12 (a, b). The previous Figure 14e is now Figure 12c, and the new Figure 12d evaluates the derived AOT40.

Figure 10, the difference between observation and model results should be more thoroughly discussed. E.g. where does the large difference in the temporal extent and timing of Vd come from?

An error in plotting the MLM data has been corrected. Additional analyses have been conducted (see additions to Table 6 and Figure 10 and the new Figure S5) to support the extended discussions on this case study in terms of the diurnal and daily variability of the fluxes.

technical corrections purely - technical corrections

P1L18: I'm missing the acronym WRF-chem.

"Weather Research and Forecasting model with online Chemistry" only appears once in the abstract, which stands alone from the main body the manuscript. Therefore, it is not necessary to define an acronym for it here. The definition of WRF-Chem in the manuscript occurs in the last paragraph of Section 1.

P5L136/L142/L148: "lass access" Typo "last"?

Corrected.

P5L157: GV F should read GVF.

Done.

P7L204/L209: The "[ ]" are not necessary.

Removed.

P7L211: ")surface" missing whitespace.
Done.

P9L274: unban Typo "urban"
Corrected.

P9L282 and others: ~three months use of ~ in text body should be avoided. Maybe use "approximately" or "roughly" or "about" instead.
Changed to "*approximately*".

P15L554: "[...] dramatically higher [...]" The authors may consider using a more neutral formulation.
Changed to "*remarkably*".

P16L515: "aggressive efforts" The authors may consider using a more neutral formulation.
Changed to "*strong*".

Figure 3/4: '-' should probably read '−'?
All figure captions and text have been extensively edited to ensure the uses of "−" and "," are consistent with the formats that ACP used in Huang et al. (2021).

**Response to RC2**

General comments - This manuscript assesses the influence of assimilating soil moisture observations on modeled ozone dry deposition using either a Wesely or dynamic scheme in two commonly used land surface models. They report a much stronger sensitivity to soil moisture within the dynamic scheme and assess how this affects health and ecological endpoints related to ozone damage. This is a novel approach that moves the science forward and fits well with the scope of ACP, although more evidence is needed for several of the conclusions. The model evaluation is not mechanistic and at times conflicts with conclusions drawn about the observational constraints. For example, biases are attributed to both the model and the soil moisture dataset in dense vegetation regions but without clear justification. There is a noticeable dearth of citations, both for previously established concepts and the datasets used, and the abstract could be sharpened to better reflect the key takeaways of this manuscript. With these major changes implemented, I believe that this paper may be ready for publication.
Thank you for the overall positive feedback and useful suggestions. Major changes have been implemented, which we believe have improved the manuscript significantly. As noted in the list of major changes in page 1 of this document, we clarified, updated, and/or discussed the model representations of SM-vegetation growth relationship, which depends on β scheme, and this affects the interpretation of SM DA results. We also improved referencing and citations in all sections of the manuscript and the SI.

Specific comments
For the abstract, please consider restructuring to introduce the main questions of the study before discussing the overall methods/findings, and to sharpen the latter part of the paragraph to clarify your findings (more specifically, after the sentence that ends with "due to the data assimilation

(DA)", which I found to be quite strong and nicely written). I found the phrases "strongly affect the quantitative results" and "wide range below 20%" to be a bit vague and a bit of a missed opportunity to express your key takeaways.

We have restructured the abstract following these suggestions and believe that the revised version better expresses our key takeaways. The abstract now starts with the main goal of "*...we quantify the impact of satellite soil moisture (SM) on model representations of this process when different dry deposition parametrizations are implemented, based on which the implications for interpreting $O_3$ air pollution levels and assessing the $O_3$ impacts on human and ecosystem health are provided*", followed by the overall methods/findings. The descriptions on "strongly affect the quantitative results" and "wide range below 20%" have been changes to "*Further, through case studies at two forested sites with different soil types and hydrological regimes, we highlight that, applying the Community Land Model-type of SM factor controlling stomatal resistance (i.e., β factor) scheme in replacement of the Noah-type β factor scheme reduced the $v_d$ sensitivity to SM changes by ~75% at one site while doubled this sensitivity at the other site*" and "*1–17%*", respectively.

Lines 43-44: Please change the word "ever-tightening" to something like "tighter." It is not a given that air quality standards will continue to be more stringent.

Changed as suggested.

Line 46: Please include more of the pertinent references here rather than only citing the companion paper.

Several references have been added.

Line 48: For the sentence that ends "deposited chemicals' concentrations", please consider citing Baublitz et al. 2020.

This study is now cited.

Lines 79-end of paragraph: Specify which of the preceding references compare Wesely parameterizations (e.g. Wong et al. 2019, Wu et al., 2018) and factor of 2 differences (e.g. Clifton et al. 2017).

This paragraph focuses on discussing uncertainty in $v_d$ calculated based on the Wesely scheme in models at multiple scales. We added "*Studies such as Hardacre et al. (2015) show that..*" before describing differences in Wesely-based $v_d$ attributable to model configurations. Almost all of the cited studies reported large model (Wesely-based)-observation discrepancies at sparsely distributed sites, and this point has been clarified. The following paragraph focuses on introducing the comparisons between different dry deposition schemes. We have also modified this paragraph to include point-scale modeling that some of the cited work was based on (e.g., Wu et al., 2018).

Line 125 – Consider mentioning that these schemes will be described in section 2.3.

Added "*(details in Section 2.3)*".

Paragraph starting at line 187 – It's not clear how this connects with the focus of your paper. Please consider cutting this paragraph or expanding on relevant connections.

As demonstrated in existing Noah-MP LSM multiphysics studies (e.g., Yang et al., 2011) and multimodel intercomparison studies, the modeled SM, surface temperature, evapotranspiration, runoff, carbon fluxes and snow (not applicable to this study) highly depend on how these physics

schemes are chosen. Therefore, it is necessary to clearly list what were implemented in this work based on our prior experiences with Noah-MP and the recommendations in literature. We added "*..which can affect the modeled land state and flux variables include..*".

281-282 – More information is needed about this extrapolation as it's not immediately clear this is justified/warranted. Is the 13-day period in the middle of the extrapolation? How are you accounting for seasonal effects on ozone? How does the seasonal cycle of vegetation (and vd) compare with the seasonal cycle of ozone, and how are you accounting for the differences?

We now provide additional information in the SI and text on seasonal surface $O_3$ concentrations (based on AQS and CASTNET observations) and surface fluxes (inferred from carbon and energy fluxes according to the correlations between carbon/energy and $O_3$ fluxes which have been shown in earlier figures and discussions), to support the discussions in Section 3.3. Seasonal conditions based on observations and observation-derived datasets have been provided for three consecutive months of April-May-June, May-June-July, June-July-August, and July-August-September of 2016 which are related to the 13-day results. Surface AOT40 and surface fluxes reached their peak values during April-May-June and June-July-August 2016, respectively. With the model biases being accounted for, the linearly scaled $POD_y$ and AOT40 values are overall underestimated referring to such seasonal variability of surface $O_3$ and fluxes. We also recognize the interannual variability in $O_3$ concentrations and fluxes that affects impact assessments based on conditions of a particular year, and this is typical in studies that estimate $O_3$ impacts based on multiscale modeling results for a selected period (e.g., Lapina et al., 2014, Figure 3 and relevant discussions). It has been made clear in text that we focus on "*qualitatively interpreting the results and discussing their implications*" here and refer to results from other studies focusing on other time periods.

Lines 316-317 – a nice finding about the wet/dry biases by forest coverage

Thanks.

Lines 323-324 – What was the r value before? What is the implication of the increase?

The *r* values before are introduced at the beginning of this section: "*They are moderately correlated with the column-averaged SM fields (r=0.875 and 0.871, respectively)*". The slight enhancements in the correlations reflect the changes of SM across the entire soil column which have been discussed in previous sentences.

Sentence starting on line 336: "The likely degraded model performance…" What evidence do you have that the SM-vegetation growth feedbacks contribute the model bias? Later (lines 352-354) it's suggested that dense vegetation challenges SM DA, but here the DA seems to be assumed to be true. Consider providing more evidence for your claim and/or providing context for DA uncertainty.

The effectiveness of DA for this application is sometimes challenged by dense vegetation and the model parameterizations including the representation of SM-vegetation growth relationship. The former has been discussed in Huang et al. (2021) and SM-vegetation growth relationship is now introduced in Figure S1 caption. The model representation of SM-vegetation relationship relies on the water stress coefficient which is sensitive to the applied β scheme in Noah-MP. The added maths formulations of β factor in Section 2.2 and the reported ΔSM−ΔGVF correlation coefficients for the Noah_D and CLM_D cases (Figure S1, right) support this statement. Additionally, the

uncertainty in satellite derived GVF data, which is now discussed in Figure S2 caption and text, affects our assessments on the model performance of vegetation.

Please provide citations for the sentence starting line 352: "The EF values were unfavorably reduced… in previous studies." What studies?
This point has been discussed in our part 1 study which is now cited here.

Paragraph starting at line 363: The SIF results are compelling and support your prior analysis. It's not as clear how the OCS component relates to your investigation. This paragraph is also oddly positioned in that it interrupts the EF discussion. Consider re-structuring it and connecting the OCS component back to previous discussion. Alternatively, cut the OCS analysis and incorporate the SIF component as a sentence or two in the previous paragraph. What is the timeframe for the ACT-AMERICA campaign? A figure caption states 2016, but this paragraph is talking about 2004 … ? I'm finding it challenging to follow this part of the analysis.
We agree that the discussions of anthropogenic, ocean, and soil interferences are a little interruptive, which have been incorporated into Figure S3. The ACT-America OCS data collected in 2016 are being discussed here, referring also to the OCS drawdowns measured during a 2004 campaign over the same regions (Campbell et al., 2008). The OCS related sentences have been reworded to be more closely linked to the previous discussion: "*All these datasets suggest moderate-to-high terrestrial carbon uptake around the Lower Mississippi croplands and the forests/croplands near the Texas-Oklahoma border, which is supported by the large OCS drawdowns (i.e., the free tropospheric-near surface gradients far exceeded 60 pptv) along with other trace gas measurements taken onboard the B-200 and C-130 aircraft*".

Lines 405-406: "results from many existing model- and measurement-based studies" needs citations.
A few citations have been added "*(e.g., Val Martin et al., 2014; Hardacre et al., 2015; Silva and Heald, 2018; Lin et al., 2019)*".

Lines 539-541 – Please expand on the evidence for your conclusion that the scaled POD and AOT40 values are underestimated.
Please refer to our response to your comment on L281-282.

Lines 621: This has been suggested in other papers, consider citing Clifton et al. 2020, He et al., 2021, Baublitz et al., 2020.
This sentence has been extended to include: "*, a point that has also been brought up in previous dry deposition modeling works (e.g., Baublitz et al., 2020; Clifton et al., 2020)*".

Section 3.1 – Consider starting this section by describing the question you aim to address here. It would be helpful to include a brief description of the simulations in the text (e.g. "Noah_D is the Noah model using the dynamic vegetation scheme").
The opening sentence of this paragraph/section now reads as: "*Figure 2 compares the horizontal and vertical gradients of the model's initial SM conditions from the Noah_D and CLM_D cases defined in Table 1, in which the Noah- and CLM-type of β factor schemes were applied*".

I found the ending to section 3.2.1 effective in drawing out the key takeaway of this analysis. I wonder if it be possible to pare down the key contributors to the DA influence on the model more (e.g. by describing the overall quantitative signal) than the somewhat broad list included here?

Excluding radiation from the list based on results in Figure 5g–l is straightforward. A sentence has been added stating that "*In many cases these primary contributing factors to the DA impacts are interdependent, and their relative contributions vary by location and time*". Note that the responses of these fields to the DA are overall (anti-)correlated. For the Noah-MP/dynamic vegetation related cases, the contributions of SM changes through β depend on the modeled SM range, soil type, and the applied β factor scheme, as also demonstrated in the following case study section, and the contributions of temperature changes depend on the modeled temperatures in the DA and no-DA cases and their distances from 25 °C, as equation (7) indicates. In the following section focusing on dry deposition fluxes, it is mentioned that, for Wesely scheme related cases, the $v_d$ differences are often largely attributable to temperature differences.

Technical corrections
Line 36: "more important role in the Earth's climate system." Can you be more specific?
We added: "*..by trapping infrared radiation and absorbing ultraviolet radiation (e.g., Lacis et al., 1990)*".

Line 67-69: Clarify who expects the impacts of SM on vd to be exacerbated or soften this claim. As written this sentence implies the IPCC makes this claim, which I don't believe is true. The following sentence is clearer.
Yes, the cited IPCC report covers droughts but not the SM impacts on $v_d$ under the warming/drying environment. This sentence has been revised to "*The SM impacts on $v_d$ and atmospheric states through the above-mentioned pathways are likely to continue to grow in future. This is because, according to Intergovernmental Panel on Climate Change (2021), the occurrence and severity of droughts, some of which are characterized by surface and/or column-averaged SM deficits, are projected to increase over many US regions under warmer future environments.....*"

Line 123 – does LIS/WRF-Chem have a version number? Line 124 – SMAP citation? Accessed date?
The SMAP SM (the mission citation has been added) data and LIS/WRF-Chem basic versions are consistent with Huang et al. (2021) to allow comparisons of results in part 1 and 2. Some necessary changes (e.g., irrigation process based on Noah-MP) are made referring to newer versions of the tool.

Lines 132-133 – citations for IGBP MRIS, other dataset?
These can now be found in note a of Table S1, which is referred to at this line.

Line 136: lass -> last
Corrected.

Line 142: citation, access date for NLCD
As suggested at the source of the NLCD (https://www.mrlc.gov/data/nlcd-2016-land-cover-conus), Wickham et al. (2021) is now cited.

252-256 – citations for these datasets?
Please see Table 2 for data attributes and references in data availability sections of this paper and Huang et al. (2021).

Line 331 – Please clarify what is meant by "The DA adjusted the modeled GVF and SLM fields toward similar directions"
A panel has been added to Figure S1, and this sentence has been changed to: "*Overall, the DA adjustments to the modeled GVF and SM fields are positively correlated (Figure S1, right), and the relative changes in GVF are smaller*". In Figure S1 caption, the model representations of the SM-vegetation dynamics relationships are introduced in detail.

Sentence starting line 349: "Larger GPP and EF values…" the phrase "most of which" – please clarify if this relates to CLM_D or Noah_D?
"Most of which" refers to larger values in CLM_D. This sentence has been changed to: "*Larger GPP and EF values are found in CLM_D than in Noah_D, most of these larger values match better with the SMAP L4C and FLUXCOM data*".

Line 383 – "skin temperature" is "surface temperature"?
These names are often interchangeable. Modified as suggested.

Sentence starting "These results can be mainly explained…" A nice sentence/finding.
Thanks.

Figure 2 - difficult to read colorbar numbers, subplots e-h tough to see color gradient
The font of these numbers has been increased and a sharper color gradient is now used for e−h.

Figure 4 caption – says (g-j) but believe it should be (f-j). Can't see colorbar numbers
(f) is not WRF-Chem based results and as introduced in "*Period-mean SMAP L4C GPP and FLUXCOM evaporative fraction are shown in (a, f)*". Horizontal colorbars are now used with enhanced readability.

Figure 8 – what is "non-urban"?
"non-urban terrestrial regions" referred to overland model grids whose LULC types do not belong to the "urban" category defined in Figure 1a. This sentence has been reworded.

Figure 9 – parts a) and b) look almost the same… consider simplifying to just $v_d$?
Both panels were shown because $v_d$ and $F_t$ have different meanings and they are both included in Figure 7, Tables 5 and 6. But yes, we agree that b) can be dropped, and a note in figure caption that the $F_t$ based results are similar to the $v_d$ based is sufficient. We now also include *r* and standard error (%) of the slopes in Figure 9 to address another comment.

Please break up the sentence starting on Line 634 for clarity, in particular of the last clause: "While the multiple no-DA…" I think that "a common issue shared…" refers to the positive O3 biases, but the way that it's written, it's not clear if referring to this or to the DA exacerbating the O3 Please also include citations, eg: Li et al., 2018; Travis & Jacob, 2019; Val Martin et al., 2014

This sentence has been separated into two. Yes, the common issue referred to the positive surface $O_3$ biases in free running global/regional modeling systems for this region/season. To suggest areas for improvement in the future, we added Li et al. (2016) on dynamically modeling $O_3$ impacts on vegetation, Jiang et al. (2018) on biogenic emissions, and Makar et al. (2017) on the reduction of photolysis reaction rates and the modification of vertical transport due to the presence of foliage.

---

## Referee Report (RR1)

This paper improves our understanding of the role of soil moisture in ozone deposition variability, while offering a new tool to assess how this connection changes into the future with climate and shifts in anthropogenic emissions. It is a good fit for ACP that moves the science forward. The sole remaining suggestion that I have is unrelated to the manuscript's conceptual basis, core arguments or evidentiary support. Acknowledging that the authors have already improved the figures in this revision, I believe there are some places where the figures would be more effective, and more accessible, if the numbers/text were made larger (for example, the right-side colorbars for Figs 2-4 and 14,15). More broadly, the authors have been thorough in addressing my concerns with the original draft, and I believe that this version is acceptable for publication in its current form.

---

## Editor Decision (ED1)

P2, L55: "climate as well" should read "as well as climate"

P2, L58: replace as "as well" by "and"

P3, L66: aboveground → above ground (space is missing)

P5, L141: "Same as in Huang et al. (2021)" should read "As in Huang et al. (2021)"

P6, L178: What is "TV"? At least once the variables should be introduced with their long names.

P7, L209: Chen97? I think here are more details needed.

P7, L215: Add "The" so that it reads "The Sprinkler scheme was……."

P7, L220: Abbreviation "GVF" not introduced.

P8, L239: Same with "PAR"

P8, L241-242: Sentence grammatically not correct. Please rephrase.

P8, L244: equations (14) → Eq. (14)

P9, L256: equations (13) and (15) → Eq. (13) and (15)

P9, L262: I would suggest to put GPP in parentheses.

P10, L281ff: Sentence too long and complicated. I would suggest to either split the sentence into two sentences or to make a bullet list.

P10, L298: equations (17) and (18) → Eq. (17) and (18)

P12, L371-372: Abbreviations L4C and FLUXCOM introduced?

P13, L392: Same here with "SIF" and "ACT"

P14, L420: equation 2 → Eq. (2)

P14, L421: Figure 5g-l → Fig. 5 g-l

P14, L430: Figure 8 → Fig. 8

P15, L471: Abbreviation "CASTNET" not introduced

P16, L479: Check sentence. Sounds grammatically not correct.

P16, L487: Abbreviation "MLM" and "GEM" not introduced.

P17, L535: "beta"?

P18, L568: EF? Abbreviation not introduced.

P19, L589: Figures → Fig. And space are missing.

P19, L599: RBL/RYL? Abbreviation not introduced.

P21, L653: Abbreviation "AQMEII4" not introduced.

Generally throughout the manuscript "or/and" should read "and/or".

---

## Author Response (AR2)

The feedback from Dr. Khosrawi and Anonymous Referee #2 is greatly appreciated, and changes have been made to the manuscript accordingly. Please see below our point-by-point response (in blue) to all comments (in black). Quoted text from the revised manuscript is *in italic*. A "tracked-changed" version of the manuscript is attached to this document.

**Response to the Editor's review**

P2, L55: "climate as well" should read "as well as climate"
This sentence has been changed to: "*..moreover, it could contribute to a more reasonable assessment of the $O_3$ impacts on vegetation (e.g., Mills et al., 2011; Lombardozzi et al., 2015; Mills et al., 2018b; Ducker et al., 2018; Ronan et al., 2020; Fu et al., 2022), which is also relevant to the budgets of other greenhouse gases, weather, and climate.*"

P2, L58: replace as "as well" by "and"
Changed as suggested.

P3, L66: aboveground → above ground (space is missing)
Changed to "*above-ground*", which is consistent with the use in Section 2.4 (L281).

P5, L141: "Same as in Huang et al. (2021)" should read "As in Huang et al. (2021)"
Changed as suggested.

P6, L178: What is "TV"? At least once the variables should be introduced with their long names.
"TV" is defined in the following sentence: "*TV, $P_{air}$, $e_{air}$, and $e_{sat}(TV)$ are canopy temperature, surface air pressure, vapor pressure at the leaf surface, and saturation vapor pressure inside leaf, respectively*".

P7, L209: Chen97? I think here are more details needed.
This has been introduced in detail in Niu et al. (2011) and Section S1 of Huang et al. (2021), which are now cited here.

P7, L215: Add "The" so that it reads "The Sprinkler scheme was……."
Changed as suggested.

P7, L220: Abbreviation "GVF" not introduced.
"GVF" is defined at L166: "*green vegetation fraction (GVF) does not come from…*".

P8, L239: Same with "PAR"
"PAR" is defined at L187-188: "*…PAR represents the photosynthetically active radiation per unit LAI*".

P8, L241-242: Sentence grammatically not correct. Please rephrase.
This sentence now reads: "*The Wesely-scheme related results that are new from this study and those from Huang et al. (2021) are compared (Table 1)*".

P8, L244: equations (14) → Eq. (14)

Changed as suggested.

P9, L256: equations (13) and (15) → Eq. (13) and (15)
Changed as suggested.

P9, L262: I would suggest to put GPP in parentheses.
We have modified this sentence to more clearly list the focused surface flux variables, and now "GPP" is in parentheses.

P10, L281ff: Sentence too long and complicated. I would suggest to either split the sentence into two sentences or to make a bullet list.
GPP from SMAP L4C and the two GPP proxies are now introduced as 2) and 3), respectively.

P10, L298: equations (17) and (18) → Eq. (17) and (18)
We changed "equation" to "Eq." throughout the paper.

P12, L371-372: Abbreviations L4C and FLUXCOM introduced?
"L4C" stands for "*level 4 carbon*", which is defined in Section 2.4 (L282). According to key references of FLUXCOM data products, FLUXCOM does not seem to have a long name.

P13, L392: Same here with "SIF" and "ACT"
"SIF" and "ACT" stand for "*solar-induced chlorophyll fluorescence*" and "*Atmospheric Carbon and Transport*", respectively, which are defined in Sections 2.4 (L284-285) and 2.2 (L176), respectively.

P14, L420: equation 2 → Eq. (2)
We changed "equation(s)" to "Eq(s)." throughout the paper.

P14, L421: Figure 5g-l → Fig. 5 g-l
P14, L430: Figure 8 → Fig. 8
We changed "Figure" to "Fig." throughout the paper, except when it appears at the beginning of a sentence or a figure/table caption.

P15, L471: Abbreviation "CASTNET" not introduced
"CASTNET" stands for "Clean Air Status and Trends Network", as first introduced in Section 2.4 (L278).

P16, L479: Check sentence. Sounds grammatically not correct.
This long sentence has been broken into two.

P16, L487: Abbreviation "MLM" and "GEM" not introduced.
"MLM" is short for "*multilayer model*", as defined in Section 2.4 (L289). "Noah-GEM" is now written as "*Noah-Gas Exchange Model*". Also note that, when discussing factual data used in the MLM calculations, we changed "conditions in the 2000s" to "*conditions in the 2000s*" for clarity.

P17, L535: "beta"?

We changed "2016beta" to "2016 beta", the latter of which is consistent with the use in Huang et al. (2021).

P18, L568: EF? Abbreviation not introduced.
"EF" stands for "evaporative fraction", which was first defined in Section 3.2.1. In the revised version it is introduced at the beginning of Section 2.4 (L264).

P19, L589: Figures → Fig. And space are missing.
We changed "Figures" to "Figs." throughout the paper, except when it appears at the beginning of a sentence or a figure/table caption. And "12(c,d)" has been changed to "*12(c, d)*".

P19, L599: RBL/RYL? Abbreviation not introduced.
"RBL" and "RYL", which are short for "*Relative Biomass Loss*" and "*Relative Yield Loss*", respectively. Please see their definitions in Section 2.4 (L306).

P21, L653: Abbreviation "AQMEII4" not introduced.
The full name of AQMEII4 is now given: "*Air Quality Model Evaluation International Initiative Phase 4*".

Generally throughout the manuscript "or/and" should read "and/or".
Changed as suggested.

**Response to Anonymous Referee #2's report**

This paper improves our understanding of the role of soil moisture in ozone deposition variability, while offering a new tool to assess how this connection changes into the future with climate and shifts in anthropogenic emissions. It is a good fit for ACP that moves the science forward. The sole remaining suggestion that I have is unrelated to the manuscript's conceptual basis, core arguments or evidentiary support. Acknowledging that the authors have already improved the figures in this revision, I believe there are some places where the figures would be more effective, and more accessible, if the numbers/text were made larger (for example, the right-side colorbars for Figs 2-4 and 14,15). More broadly, the authors have been thorough in addressing my concerns with the original draft, and I believe that this version is acceptable for publication in its current form.
Thank you for the overall positive feedback. We have revised Figs. 2-4, 14, 15, and S2 according to this comment.

[revised manuscript text omitted]